# Knowledge-Informed Deep Learning for Hydrological Model Calibration: An Application to Coal Creek Watershed in Colorado

Peishi Jiang[1], Pin Shuai[1,2], Alexander Sun[3], Maruti K. Mudunuru[1], and Xingyuan Chen[1]

[1]Atmospheric Sciences and Global Change Division, Pacific Northwest National Laboratory, Richland, Washington, USA
[2]Department of Civil and Environmental Engineering, Utah Water Research Laboratory, Utah State University, Logan, Utah, USA
[3]Bureau of Economic Geology, Jackson School of Geosciences, The University of Texas at Austin, Austin, Texas, USA
**Correspondence:** Peishi Jiang (Peishi.Jiang@pnnl.gov)

**Abstract.** Deep learning (DL)-assisted inverse mapping has shown promise in hydrological model calibration by directly estimating parameters from observations. However, the increasing computational demand for running the state-of-the-art hydrological model limits sufficient ensemble runs for its calibration. In this work, we present a novel knowledge-informed deep learning method that can efficiently conduct the calibration using a few hundred realizations. The method involves two steps. First, we determine decisive model parameters from a complete parameter set based on the mutual information (MI) between model responses and each parameter computed by a limited number of realizations ($\sim$50). Second, we perform more ensemble runs (e.g., several hundred) to generate the training sets for the inverse mapping, which selects informative model responses for estimating each parameter using MI-based parameter sensitivity. We applied this new DL-based method to calibrate a process-based integrated hydrological model, the Advanced Terrestrial Simulator (ATS), at Coal Creek Watershed, CO. The calibration is performed against observed stream discharge (Q) and remotely sensed evapotranspiration (ET) from the water year 2017 to 2019. Preliminary MI analysis on 50 realizations resulted in a down-selection of seven out of fourteen ATS model parameters. Then, we performed a complete MI analysis on 396 realizations and constructed the inverse mapping from informative responses to each of the selected parameters using a deep neural network. Compared with calibration using observations covering all time steps, the new inverse mapping improves parameter estimations, thus enhancing the performance of ATS forward model runs. The Nash-Sutcliffe efficiency (NSE) of streamflow predictions increases from 0.53 to 0.8 when calibrating against Q alone. Using ET observation, on the other hand, does not show much improvement on the performance of ATS modeling mainly due to both the uncertainty of the remotely sensed product and the insufficient coverage of the model ET ensemble in capturing the observation. By using observed Q only, we further performed a multi-year analysis and show that Q is best simulated (NSE > 0.8) by including in calibration the dry year flow dynamics that show more sensitivity to subsurface characteristics than the other wet years. Moreover, when continuing the forward runs till the end of 2021, the calibrated models show similar simulation performances during this evaluation period as the calibration period, demonstrating the ability of the estimated parameters in capturing climate sensitivity. Our success highlights the importance of leveraging data-driven knowledge in DL-assisted hydrological model calibration.

# 1 Introduction

Calibrating a hydrological model is critical to accurately capturing the hydrological dynamics of the simulated watershed, which in turn improves the understanding of the corresponding terrestrial water cycle (Singh and Frevert, 2002). While the increasing complexity and spatio-temporal resolution of the hydrological models enable a better representation of the watershed dynamics (Kollet and Maxwell, 2006; Coon et al., 2019; Wang and Kumar, 2022), running these models is computationally expensive (Clark et al., 2017) even with existing high-performance computing resources. This computational burden significantly impedes the efficient and accurate calibration of integrated hydrological models.

Balancing the trade-off between computational cost and calibration accuracy is necessary when adopting traditional model calibration methods (Kavetski et al., 2018). Newton-type optimization methods (Jorge and Stephen, 2006; Qin et al., 2018) are known for their fast iteration convergence but usually only achieve local optimum. On the other hand, the stochastic methods, such as the shuffled complex evolution algorithm (Duan et al., 1992), the dynamically dimensioned search algorithm (Tolson and Shoemaker, 2007), and the ensemble Kalman filter (Reichle et al., 2002; Moradkhani et al., 2005; Evensen, 2009; Sun and Sun, 2015), are capable of providing better global optimum at the cost of high computational demand. One alternative is to use a surrogate model that provides fast emulations to replace the physical model during calibration so that one might save the computational budget while achieving a reasonable calibration result. Mo et al. (2019) employed a dense convolutional encoder-decoder network as the emulator for a two-dimensional contaminant transport model to estimate the conductivity field using iterative local updating ensemble smoother. Similar subsurface characterization was also performed by Wang et al. (2021) who developed a theory-guided neural network as the surrogate of a flow model which was coupled with an iterative ensemble smoother to estimate the subsurface characteristics. In light of the dimensionality reduction of the model states, Dagon et al. (2020) calibrated biophysical parameters using a global optimizer on a surrogate that emulates the principle components of the outputs of community land models. Jiang and Durlofsky (2021) adopt a recurrent encoder-decoder network as the data-space inversion parameterization to reduce the dimensionality of the model states/parameters and used ensemble smoother with multiple data assimilation to update the low-dimension latent variables. Despite the successes of using surrogates in calibration, how to develop an accurate and trustworthy emulator can vary from case to case and, in fact, is still a long-standing challenge (McGovern et al., 2022).

Recently, Deep Learning (DL)-assisted inverse mapping shows promise in addressing inverse problems and has seen early successes in hydrology (Cromwell et al., 2021; Mudunuru et al., 2021; Tsai et al., 2021), petroleum engineering (Razak et al., 2021), and geophysics (Yang and Ma, 2019; Wang et al., 2022). By employing a well-trained DL model (Goodfellow et al., 2016), this approach maps model parameters from model states/outputs/responses such that once trained, the mapping can directly infer the parameters based on observations. The inverse mapping outperforms the traditional calibration approaches in the following ways. First, DL models can better capture the highly nonlinear relationships encoded in the model than ensemble-based methods, which primarily rely on the linear estimation theory through the Kalman filter (Evensen, 2009; Moradkhani et al., 2005; Reichle et al., 2002; Sun and Sun, 2015). Yang and Ma (2019) developed a convolutional neural network-based inverse mapping that outperforms the traditional full waveform inversion adopting the adjoint-state optimization method in

estimating seismic velocity from seismic data. Cromwell et al. (2021) also demonstrate the improved performance of DL-assisted inverse mapping over ensemble smoother in estimating subsurface permeability used in a watershed model based on the Advanced Terrestrial Simulator (ATS). Second, training DL models may potentially use fewer realizations than the traditional methods such as iterative calibration methods that usually require several thousands of realizations to achieve the model optimization convergence. Mudunuru et al. (2021) show that DL-assisted inverse mapping using 1,000 realizations outperforms dynamically dimensioned search algorithm (Tolson and Shoemaker, 2007) that has to leverage 5,000 realizations in calibrating multivariate parameters of models based on the Soil and Water Assessment Tool (SWAT). Third, the calibration workflow is simpler given that ensemble simulations don't have to be fully coupled with the inverse mapping. Traditional calibration methods require sophisticated workflows (White et al., 2020; Jiang et al., 2021) to manipulate the model restart (e.g., ensemble Kalman filter), model rerunning (e.g., gradient-based and ensemble-based methods), and the communications between hydrological model and calibration tools, which can be time-consuming. Meanwhile, such an integrated workflow tool is not necessary for developing inverse mapping because model ensemble runs and DL training are now two separate steps. This decoupling of ensemble runs and DL training allows us to use high-performance computing resources to calibrate hydrological models efficiently.

Despite its success, the current DL-assisted inverse mapping is often designed to take all observed states in estimating hydrological model parameters. However, some observational values can be uninformative, or even misinformative, to estimate parameters (Loritz et al., 2018), thus impeding the mapping performance. While the underlying assumption is that the trained DL model can 'automatically' delineate the accurate relationship between parameters and observed responses, the limited realizations (e.g., a few hundred) would potentially restrain the DL model from being well trained (Moghaddam et al., 2020). Further, when using all observed responses as inputs, the potentially large amount of trainable weights of the DL model can make the model training hard and induce the overfitting of the model (Ying, 2019), thus calling for more realizations used in training. Lately, several studies proposed new inverse mapping methods that indirectly address this issue by using dimensionality reduction and differential programming. Razak et al. (2021) developed a latent-space inversion that performs the inverse mapping from the model responses to parameters in their reduced spaces through an autoencoder. The dimensionality reduction by using an autoencoder is supposed to not only lower the original high-dimensional data space but also indirectly distill the most relevant dynamics to the parameters. Tsai et al. (2021) leveraged differential programming such that the loss function of an inverse mapping is designed to directly minimize the difference between observations and the responses predicted by a differentiable version of the physical model using the estimated parameters. In doing so, the uninformative responses are automatically given less importance in the loss. Nevertheless, both studies reduce the irrelevant information in implicit ways through complicating the DL-based mapping which, due to their 'black-box' nature, does not explicitly show to what extent an observation is relevant to a parameter. Also, by adding another layer of complexity, the inverse mapping can potentially be hard to build. For instance, the current solutions to develop a differentiable physical model rely on either a surrogate or rewriting the model using differentiable parameters (Karniadakis et al., 2021), both of which are research challenges that go beyond addressing the inverse problem itself.

The emergence of knowledge-informed DL provides a new opportunity to resolve the uninformative or misinformative issue by explicitly encoding the complex relationship between the inputs and outputs in the DL model (Willard et al., 2020). Knowledge-informed DL includes, but is not limited to, the following three ways: (1) physics-guided loss function, (2) hybrid modeling, and (3) physics-guided design of architecture. Physics-guided loss function embeds the mathematical relation between inputs and outputs in the loss function, known as physics-informed deep learning (Karniadakis et al., 2021), and has seen some early successes in earth science. For instance, Jia et al. (2019) leveraged an energy conservation loss in developing a physics-guided recurrent neural network to simulate lake temperature. Hybrid modeling, on the other hand, directly integrates the physical model with the DL model, which often serves as a surrogate for its computationally intense counterpart in the physical model (Kurz, 2021). An example can be coupling a DL-based emulator for turbulent heat fluxes with a process-based hydrological model framework (Bennett and Nijssen, 2021). Lastly, the physics-guided design of architecture explicitly designs the neural networks consistently with prior knowledge. The widely-used convolutional (Atlas et al., 1987) and recurrent (Rumelhart et al., 1986) neural networks fall into this category due to their specific network structures to learn the spatial and temporal relationships, respectively. Other related studies include relating intermediate physical variables to hidden neurons (Daw et al., 2020), explicitly learning nonlinear dynamics through the neural operator (Kovachki et al., 2021), and encoding domain knowledge obtained from nonparametric physics-based kernels into the neural network (Sadoughi and Hu, 2019). Compared with the other two types of knowledge-informed DL, which are usually limited to particular physical dynamics, the physics-guided design of architecture is more generic regarding both the processes of gaining prior knowledge and designing a correspondent neural network.

One important piece of domain knowledge is the pairwise relation between model parameters and responses. That is, how relevant a parameter is to a model response at a given time step. Understanding such a pairwise relationship is essential to select the most relevant model responses to estimate each parameter when building the inverse mapping. To this end, global sensitivity analysis (Razavi and Gupta, 2015; Sarrazin et al., 2016) is a suitable tool due to its capability to quantify the contribution of uncertainty from model inputs and parameters to model outputs, and has been extensively applied in earth system modeling (Hall et al., 2009; Harper et al., 2011; Anderson et al., 2014; Guse et al., 2014; Dai et al., 2017). Through a sensitivity analysis study on SWAT modeling (Jiang et al., 2022), mutual information (MI; Cover and Thomas (2006)) has shown the promise of using a few hundred model realizations to provide similar sensitivity results as the popular Sobol sensitivity analysis (Sobol, 2001) that usually relies on several thousand realizations. As a result, MI is well suited to unravel the relation between model parameters and responses given a few hundred realizations of a state-of-the-art fully-integrated hydrological model.

This study aims to develop a novel knowledge-informed DL method for model calibration by using a few hundred realizations. We leverage MI-based global sensitivity analysis to uncover the dependencies between parameters and observed responses, which are then used to guide the selection of crucial responses as the inputs of DL-assisted inverse mapping. We applied this method in estimating multiple parameters of a fully integrated hydrological model, ATS (Coon et al., 2019), at the Coal Creek watershed, a snow-dominated alpine basin located in Colorado, US. Multiple water years of hydrological observations are used in the both ATS model calibration and evaluation. We further performed a multi-year analysis to investigate the

significance of wet and dry years in model calibration. Our study highlights the importance of domain knowledge in uncovering the dependencies among variables of interest before hydrological model calibration.

## 2    Methods

### 2.1    Study site

The Coal Creek watershed is located in the western part of the larger East Taylor Watershed in Colorado (Figure 1(a)). The majority of the discharge flows through Coal Creek from the west to the east. The watershed is a HUC12 (Hydrologic Unit Code) watershed encompassing around 53.2 km$^2$ of the drainage area (HUC12 ID, 140200010204). According to the Köppen classification system (Köppen and Geiger, 1930), this high alpine watershed is classified as warm summer and humid continental climate with a significant snow process dominating the hydrological cycle. Based on the long-term Daymet forcing dataset (Thornton et al., 2021), the watershed receives ∼530 mm of snowfall annually, dominating its annual precipitation (∼850 mm). This watershed exhibits strong variations in topography with elevations ranging from 2706m to 3770m, where the primary land covers are evergreen forest (62.6%) and shrub (20.5%). Hydrological observations are available through (1) a USGS gaging station (station number 09111250) that records daily discharge (Q) observations at the watershed outlet; and (2) a remote sensing product of the Moderate Resolution Imaging Spectroradiometer (MODIS) 8-day composite evapotranspiration (ET) at a 500 m resolution. Figure 1 shows the time series of Q and watershed-averaged MODIS ET during the 1 October 2016 to 31 December 2021, which are used as observations for calibrating and evaluating the ATS model.

### 2.2    ATS model setup

ATS is a fully distributed hydrologic model that integrates surface and subsurface flow dynamics (Coon et al., 2019). The surface hydrological process is characterized by a two-dimensional diffusion wave approximation of the Saint-Venant equation. A three-dimensional Richards equation is used to represent the subsurface flow. The model adopts the Priestley Taylor equation to simulate evapotranspiration (ET) from various processes (e.g., snow and plant transpiration), which are coupled with the surface-subsurface hydrological cycle.

We leveraged an existing ATS setup at the Coal Creek watershed (Shuai et al., 2022). The Watershed Workflow package (Coon and Shuai, 2022) was used to delineate the mesh, the surface land covers, and the subsurface characteristics of the watershed. The resulting mesh consists of 171760 cells, formed by a two-dimensional triangle surface mesh followed by 19 terrain-following subsurface layers (Figure 1(a)). The surface mesh contains 8588 triangular cells with varying sizes that range from ∼5000m$^2$ near the stream network to ∼50000m$^2$ away from the stream network. On the surface, the National Land Cover Database was used to delineate the land cover types. In the subsurface, the 19 layers add up to 28m and contain: (1) 6 soil layers in the top 2m, (2) 12 geological layers in the middle, and (3) 1 bedrock layer in the bottom of the simulation domain. The maximum depth to bedrock (28m) was determined by SoilGrids (Shangguan et al., 2017). The subsurface characteristics of the soil and geological layers are retrieved from the National Resources Conservation Service (NRCS) Soil Survey Geo-

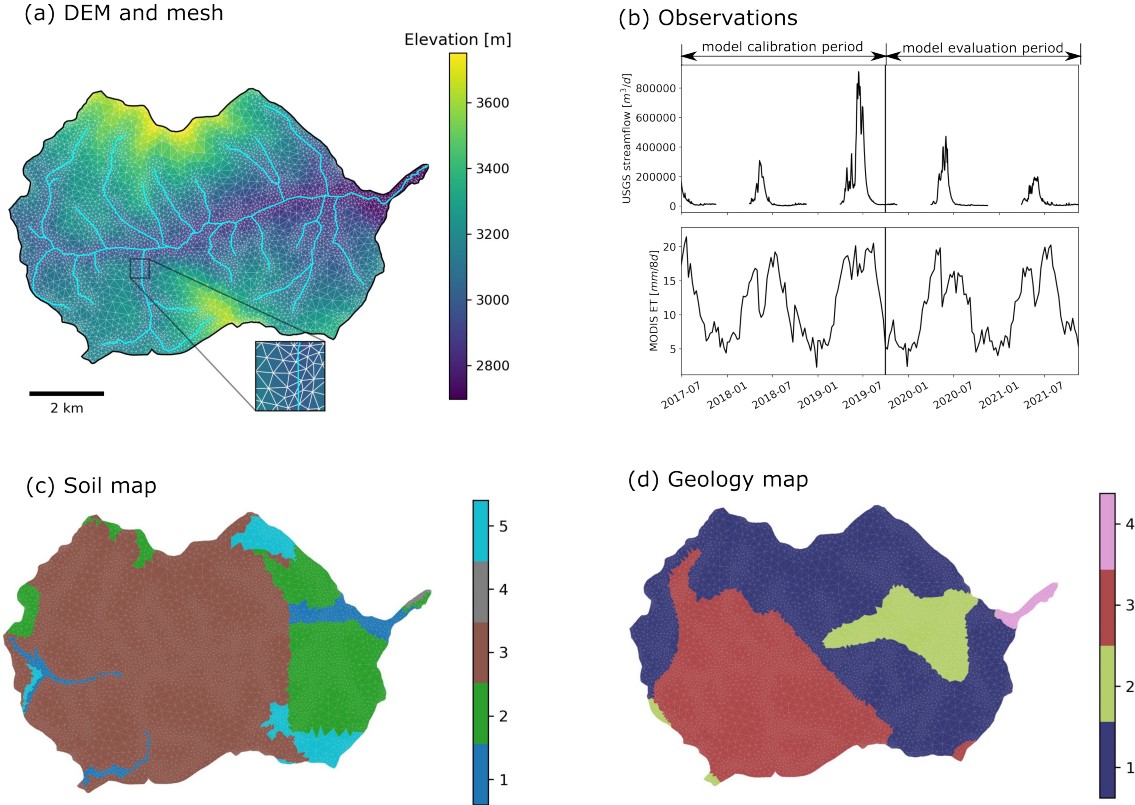

**Figure 1.** The Coal Creek watershed and the setup of the Advanced Terrestrial Simulator (ATS). (a) the river network, the digital elevation model (DEM), and the surface mesh of the watershed. (b) the time series of USGS streamflow observations (station number 09111250) at the watershed outlet and the Moderate Resolution Imaging Spectroradiometer (MODIS) 8-day composite evapotranspiration (ET) averaged across the watershed, where the observations before 1 October 2019 are used for model calibration and the remaining observations till 31 December 2021 are used for evaluating the climate sensitivity of the estimated model parameters. (c) and (d) the delineated soil and geological layers, respectively.

graphic (SSURGO) soils database and GLobal HYdrogeology MaPS (GLHYMPS) 2.0 (Huscroft et al., 2018), respectively. The k-means clustering algorithm (Likas et al., 2003) was used to group the soil and geological types based on the default per-

meability values from SSURGO and GLHYMPS, which leads to five soil types and four geological types shown in Figures 1(c) and (d), respectively. Each clustered soil or geological type is associated with a specific set of subsurface characteristics (such as permeability), which are assigned to the corresponding grouped grid cells. These subsurface characteristics are important in controlling flow dynamics and can be estimated from hydrological observations. To ensure that the model achieved a physically appropriate initial state, two spinups were performed sequentially, including (1) a cold spinup that ran the model for 1000 years

by using constant rainfall and led to steady-state model outputs (e.g., converged total amount of subsurface water storage) and (2) a warm spinup that was initialized by the steady-state spinup result and performed a transient simulation for 10 years (i.e.,

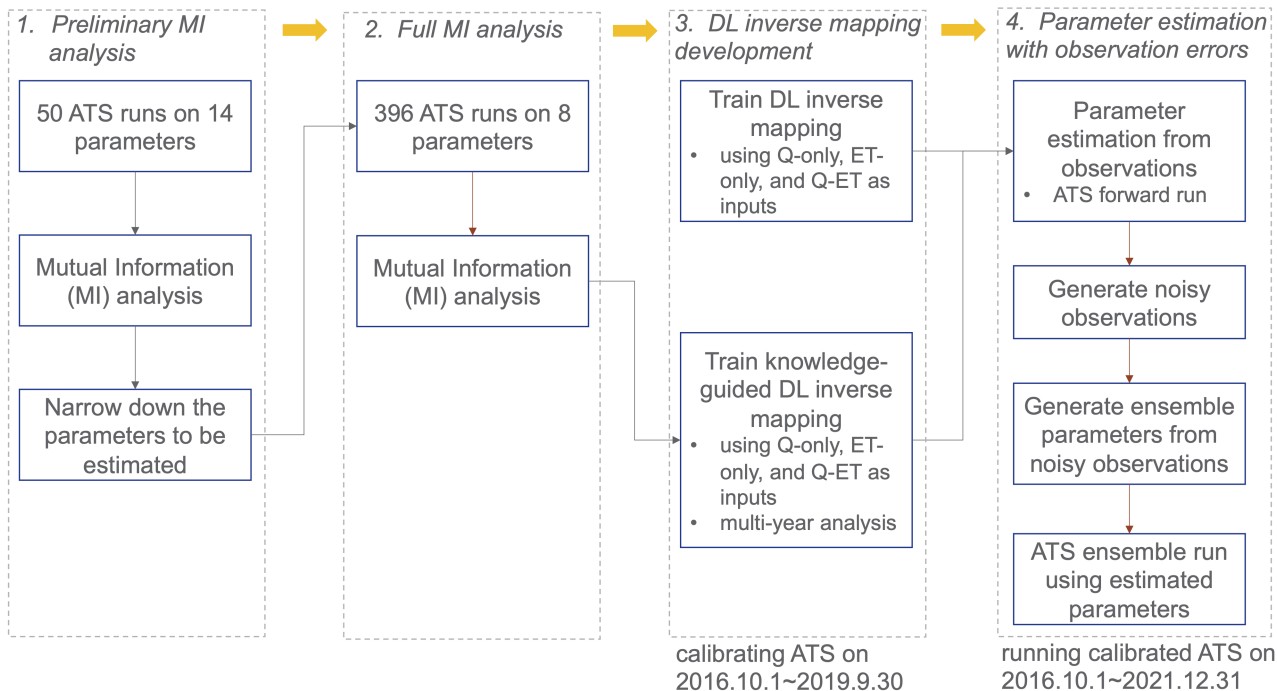

**Figure 2.** Diagram of deep learning (DL) inverse mapping development including four steps: (1) performing a preliminary mutual information (MI) analysis using 50 model runs to narrow down the parameters to be estimated; (2) performing a full MI analysis on 396 model runs to correctly delineate the sensitivity between each parameter and each observed response; (3) developing DL inverse mappings with and without being knowledge informed; and (4) estimating parameters from observations with and without observation errors.

1 October 2004 – 1 October 2014) under the Daymet forcing. Please refer to Shuai et al. (2022) for the detailed model setup and spinup.

  We select a preliminary set of 14 model parameters to be calibrated, which can be categorized into ET, snow, river channel,
170 and subsurface characteristics. The ET parameters include two coefficients used by the Priestley Taylor equation (Priestley and Taylor, 1972) in calculating the potential ET of snow and transpiration, respectively (i.e., priestley_taylor_alpha-snow and priestley_taylor_alpha-transpiration). The snow parameters are the snow melting rate (snowmelt_rate) and the temperature determining the snow melting (snowmelt_degree_diff). The river channel characteristic is the manning's coefficient (manning_n), which describes the roughness of the surface channel. The subsurface characteristics include the major soil and geological
175 permeability (i.e., perm_s1, perm_s2, perm_s3, perm_s4, perm_s5, perm_g1, perm_g2, perm_g3, and perm_g4). A detailed description of these parameters can be found in Table A1.

## 2.3 Knowledge-informed model calibration using deep learning

We develop a new methodology to calibrate ATS using knowledge-informed DL, as shown in Figure 2. The key idea is to leverage a data-driven approach to identify the sensitive model response as the inputs to the DL-assisted inverse mapping for estimating each parameter. Here, we use the MI as the sensitivity analysis tool due to its capability to uncover nonlinear relationships. Derived from Shannon's entropy (Cover and Thomas, 2006), MI quantifies the shared information between two variables: a model response $Y$ and a model parameter $X$ as follows:

$$I(X;Y) = H(Y) - H(Y|X) = \sum_{X=x} \sum_{Y=y} p(x,y) \log \left( \frac{p(x,y)}{p(x)p(y)} \right), \tag{1}$$

where $p$ is the probability density function and can be estimated by the fixed binning method; $H(Y) = -\sum_{Y=y} p(y) \log(p(y))$ is Shannon's entropy describing the overall uncertainty of $Y$; and $H(Y|X)$ is the conditional entropy that quantifies the uncertainty of $Y$ given the knowledge of $X$. Eq.(1) shows that $I(X;Y)$ is quantified as the shared dependency between the variables and is zero when $X$ and $Y$ are statistically independent. Jiang et al. (2022) show that MI computed by a few hundred realizations with a statistical significance test (SST) can yield comparable sensitivity results with the full Sobol sensitivity analysis that usually uses thousands of realizations through a multivariate sensitivity analysis of SWAT. Therefore, MI is an ideal tool to perform the sensitivity analysis on the several hundred realizations, which are relatively affordable by the computationally-intense ATS model (Cromwell et al., 2021). In this study, we follow a similar strategy of Jiang et al. (2022) to estimate $p$ using 10 evenly divided bins along each dimension and perform SST tests to filter out any non-significant MI value with a significance level of 95% based on 100 bootstrap samples. In other words, the computed MI is set to zero if the statistical significance test fails.

By using MI-based sensitivity analysis, our calibration method involves the following four steps (see Figure 2): (1) narrowing down the parameters to be calibrated using a preliminary MI analysis; (2) computing the parameter sensitivity using a full MI analysis; (3) developing knowledge-guided DL inverse mapping; and (4) parameter estimation with observational error. The details of each step are as follows:

**Step 1: Narrowing down the parameters to be calibrated.** The objective of this step is to further reduce the computational requirement by identifying the parameters that are most relevant to the responses through a coarse-resolution sensitivity analysis. To this end, we first perform a preliminary MI analysis on a sizeable preliminary parameter set. MI is computed between the model response at each time step and each parameter based on Eq.(1). Rather than getting an accurate sensitivity result, this preliminary analysis aims to provide an overview of parameter sensitivity and thus is performed on a small number of realizations to save computational cost. At Coal Creek, we generated 50 realizations based on a total of 14 parameters listed in Table A1 and performed the corresponding ATS runs to compute MI. Sobol sequence (Sobol, 1967), a quasi-Monte Carlo method, is used to generate the parameter ensemble to ensure uniform coverage of the parameter ranges. This preliminary MI analysis would allow filtering out the parameters that show little sensitivity to the model responses, thus reducing the number of parameters to be calibrated. This filtering process is performed based on whether a parameter demonstrates sufficient sensi-

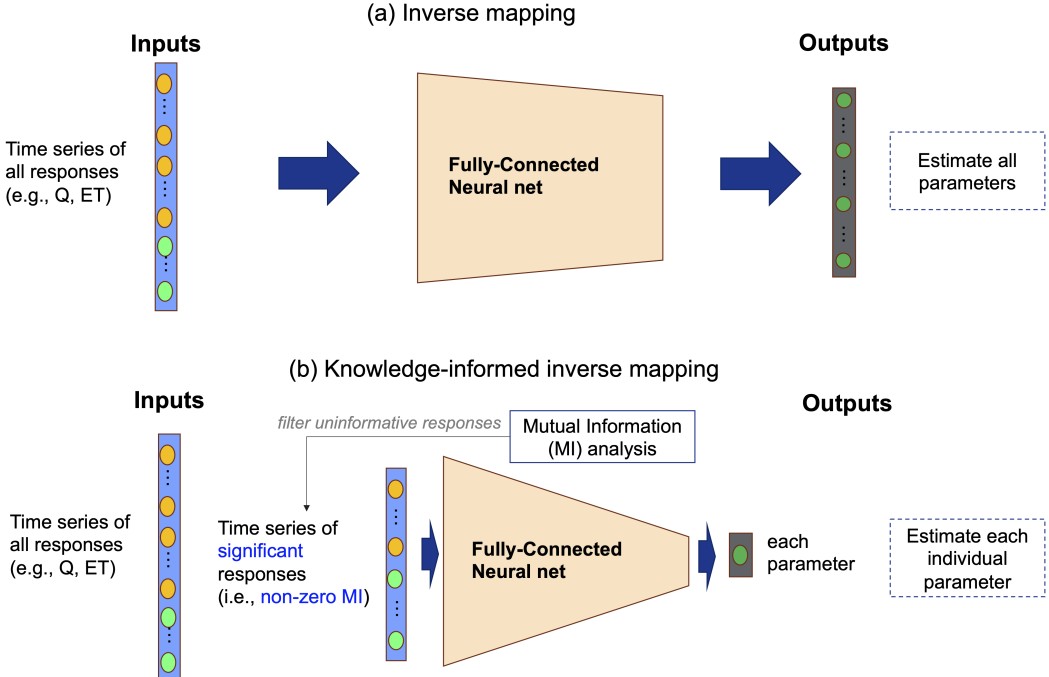

**Figure 3.** Illustration of two inverse mappings. While inverse mapping (a) estimates all parameters from all observed responses using a deep learning model (e.g., fully-connected neural net) (Cromwell et al., 2021; Mudunuru et al., 2021), the proposed knowledge-informed inverse mapping (b) estimates each parameter using only responses that shares significant MI with the parameter.

tivity across the simulation period. In this study, we selected the parameter whose proportion of the non-zero MI is larger than 5% of the overall time steps for the following full sensitivity analysis.

**Step 2: Computing the parameter sensitivity.** We then conduct a complete MI-based sensitivity analysis on the prescreened parameters in Step 1 by using a more significant number of realizations. The analysis accurately quantifies the sensitivity of model responses to each parameter, thereby uncovering their dependency. The analysis also verifies whether the parameters prescreened in Step 1 are 'truly' sensitive and can be further excluded if insensitive. At this watershed, we generated 400 ATS realizations by sampling from a reduced parameter space at Step 1 using the Sobol sequence, among which 396 ATS runs completed and are used for computing MI.

**Step 3: Developing knowledge-informed DL inverse mapping.** In this step, we develop the DL-based inverse mapping by leveraging the knowledge about the sensitivities between model responses and parameters, learned from the complete MI analysis in Step 2. While the original inverse mapping directly estimates the model parameters from the model responses using a tuned DL model (Figure 3(a)), the complete sensitivity analysis in Step 2 enables the selection of essential responses that are sensitive to a given parameter with non-zero MI as the inputs of the inverse mapping shown in Figure 3(b). Consequently, this 'filtering' step is expected to improve the performance of inverse mapping in parameter estimation by reducing the impact of noisy or unimportant model responses.

Table 1: All inverse mappings with and without being knowledged informed developed for calibrating the Advanced Terrestrial Simulator (ATS) at the Coal Creek watershed.

| Inverse mapping | Response input | Input years | Guided by MI result? |
|---|---|---|---|
| qonly-3yrs | Q | WY2017-2019 | No |
| mi-qonly-3yrs | Q | WY2017-2019 | Yes |
| etonly-3yrs | ET | WY2017-2019 | No |
| mi-etonly-3yrs | ET | WY2017-2019 | Yes |
| qet-3yrs | Q, ET | WY2017-2019 | No |
| mi-qet-3yrs | Q, ET | WY2017-2019 | Yes |
| mi-qonly-1yr-1 | Q | WY2017 | Yes |
| mi-qonly-1yr-2 | Q | WY2018 | Yes |
| mi-qonly-1yr-3 | Q | WY2019 | Yes |
| mi-qonly-2yrs-12 | Q | WY2017/2018 | Yes |
| mi-qonly-2yrs-13 | Q | WY2017/2019 | Yes |
| mi-qonly-2yrs-23 | Q | WY2018/2019 | Yes |

**Step 4: Parameter estimation with observation errors.** Once an inverse mapping is trained, we first estimate the parameters using the corresponding observations and then use the estimated parameters to perform an ATS simulation. We evaluate the calibration results using different inverse mappings by comparing the simulated responses with the observations. We further perform uncertainty analysis to analyze how sensitive the DL-based parameter estimation is to the observation errors. To this end, we generate 100 realizations of noisy observations, denoted as $\mathbf{o}_n$, such that $\mathbf{o}_n = \mathbf{o} + \epsilon \times \mathbf{o} \times \mathbf{r}$, where $\mathbf{o}$ is the vector of the original observations, $\mathbf{r}$ is the random vector with the same size as $\mathbf{o}$ and is drawn from a standard normal distribution, and $\epsilon$ is the standard deviation of the random vector $\mathbf{r}$ and is usually taken as 1/3 of a given observation error. Following Cromwell et al. (2021), $\epsilon$ is set to 0.0166 for a 5% observation error in this study. Given an inverse mapping, these noisy observations are used to estimate the ensemble parameter estimations and perform the corresponding ATS ensemble simulation.

Here, we separate the entire observations in Figure 1(b) into model calibration and evaluation periods in order to assess the adaptability of the estimated parameters to an uncalibrated period. To this end, we calibrate ATS only using the simulations during water year 2017 to water year 2019 and used the remaining observations (till 31 December 2021) for model evaluation. The ensemble runs used for sensitivity analysis and inverse modeling are performed during the calibration period. The calibrated ATS forward runs were then performed on both periods and compared against the observations in Figure 1(b). We assess the performances of the calibrated models on both periods by using two scale-independent metrics: the Nash-Sutcliffe Efficiency (NSE; Nash and Sutcliffe (1970)) and the modified Kling-Gupta Efficiency (mKGE; Kling et al. (2012)).

## 2.4 Deep learning-based inverse mapping development

For comparison purposes, we developed both the original inverse mapping and our proposed knowledge-informed version for parameter estimation. While a separate neural network is developed for estimating each parameter by using knowledge-informed inverse mapping (Figure 3(b)), the original inverse mapping estimates all parameters using one neural network and is developed by following the same strategy in Cromwell et al. (2021) and Mudunuru et al. (2021) (Figure 3(a)). Further, to assess the impact of different responses in calibration, we developed three types of inverse mappings that take various model

responses: (1) using both Q and ET; (2) using only Q; and (3) using only ET. Note that in the case of knowledge-informed inverse mapping using both observations, we take the union of the Q and ET that have non-zero mutual information as the inputs of the neural network. Additionally, a multi-year analysis was performed by training inverse mappings using Q of different combinations of observed years to evaluate both the impacts of the dry versus wet years and the number of observed years used in calibration.

All the inverse mappings developed in this study are listed in Table 1. Each mapping was developed using a multilayer perceptron (MLP) model as follows. The input of an MLP is an array concatenating the responses to be assimilated within a given calibration period. The output is the model parameter(s). Let's denote the number of input neurons, output neurons, and hidden layers as $N_i$, $N_o$, and $N_l$, respectively. $N_i$ depends on the type of inverse mapping (with or without being knowledge guided), the selections of the response variable(s), and the number of calibration years, varying from ~100 using one year of Q

to 1,785 using all three years of Q and ET. $N_o$ equals either one (i.e., estimating each parameter using knowledge-informed DL calibration) or the number of all the parameters (i.e., using inverse mapping without mutual information). Given $N_i$, $N_o$, and $N_l$, we adopt the arithmetic sequence to determine the number of neurons at each hidden layer $N_{h,l} = \lfloor N_i - \frac{N_i - N_o}{N_l} \times l \rfloor$ (where $1 \le l \le N_l$ and $\lfloor \bullet \rfloor$ is the floor function). In doing so, the information from a sequence of observed responses can be gradually propagated to estimate the parameters. We use the leaky ReLu as the nonlinear activation at the end of each layer. Based on the

order of the Sobol sequences, we sequentially split the 396 realizations into 300/50/46 for train/validation/test sets, respectively, such that each set is able to cover the full range of the parameter ensemble as much as possible. We trained each MLP using mean square error (MSE) as the loss function over 1,000 epochs with a batch size of 32. The Adam optimization algorithm, a stochastic gradient descent approach, was used to train the neural network. We performed hyperparameter tuning on each MLP using grid search to find the optimal result by varying the number of hidden layers $N_l = [1, 3, 5, 7, 9, 10]$ and the learning

rate $l_r = [1e-5, 1e-4, 1e-3]$. The performances of these mappings are further evaluated on the two magnitude-independent metrics, NSE and mKGE. To have consistent comparisons between mappings with and without being knowledge guided, both metrics are computed for the estimation of each parameter based on the test dataset (note that while the original mapping estimate all seven parameters as a whole during the training, we calculate the two metrics for each parameter separately during the postprocessing).

## 3 Results and Discussions

In this section, we present the ATS calibration result at the Coal Creek watershed using the proposed knowledge-informed DL described in Sec. 2.3. We first demonstrate the MI-based sensitivity analysis result, which is facilitated by a preliminary analysis using fewer realizations to narrow down the parameters to be calibrated. Then, we report the result of the trained DL-assisted inverse mapping performance using the simulations from the three-year calibration period on the test dataset. By using
Q alone, our result shows (1) the improved ATS simulation calibrated by knowledge-informed DL inverse mapping over the traditional mapping; and (2) the consistent performances of model forward runs between the calibration and evaluation periods. However, we also identify the ET observation extrapolation issue that impedes the calibration performance. Last, we present the multi-year analysis using observed Q only to estimate the parameters regarding (1) the performance of the simulated Q driven by the estimated parameters; and (2) the impact of the observational error on the parameter estimations using inverse
mapping.

### 3.1 MI-based sensitivity analysis

**Parameter prescreening using a preliminary MI analysis.** We calculated the MI for each daily Q/ET with each of the parameters listed in Table A1. Figure 4 shows the estimation of MI by using the 50 ATS runs in the water year 2017 (note that 20 ATS runs didn't complete after the end of the water year 2017). We then ranked the parameters based on the
285 temporally-averaged MI across the analyzed time period, as plotted on the right panel of each MI heatmap. The MI heatmaps show the varying sensitivities of different parameters over time. For instance, while Q (Figure 4(a)) is mainly sensitive to the permeability (e.g., perm_g1) during low flow seasons, snow melting parameters (e.g., snowmelt_degree_diff) play more critical roles in high flow periods than others. On the other hand, ET (Figure 4(b)) is mostly controlled by the two Priestley Taylor alpha coefficients in low and high flows, respectively. Based on the proportion of nonzero MI over all the time
steps (see Figure A1 in the appendix), we find that Q is mostly sensitive to (using a threshold of 5%) perm_s3, perm_s4, perm_g1, perm_g4, snowmelt_rate, snowmelt_degree_diff, and priestley_taylor_alpha_transpiration, and ET is mostly sensitive to priestley_taylor_alpha_transpiration, priestley_taylor_alpha_snow, perm_s3, perm_g1, and perm_g4. Consequently, we narrow down the parameters to be calibrated by taking the union of the two sets of parameters that show sensitivities to either Q or ET (also highlighted in Table A1).
**The full MI analysis.** We performed the ATS ensemble simulations on 400 realizations of the reduced parameter set and computed the MI heatmaps on the completed 396 realizations during the calibration period (i.e., water years 2017-2019) shown in Figure 5. By using more realizations, this complete MI analysis shows a better delineation of parameter sensitivity than the preliminary analysis due to its convergence on MI estimation (see the convergence of the parameter rankings in Figure A2). The convergence on a few hundred realizations is consistent with another MI-based sensitivity analysis study using Soil &
Water Assessment Tool (SWAT) (Jiang et al., 2022). Further, the MI-based parameter ranking suggests that compared with the preliminary analysis, the full analysis (1) improves the MI estimations (e.g., perm_s3); and (2) identifies the insensitive parameters (e.g., perm_s4) that are falsely considered sensitive due to the limited samples in the preliminary analysis (see

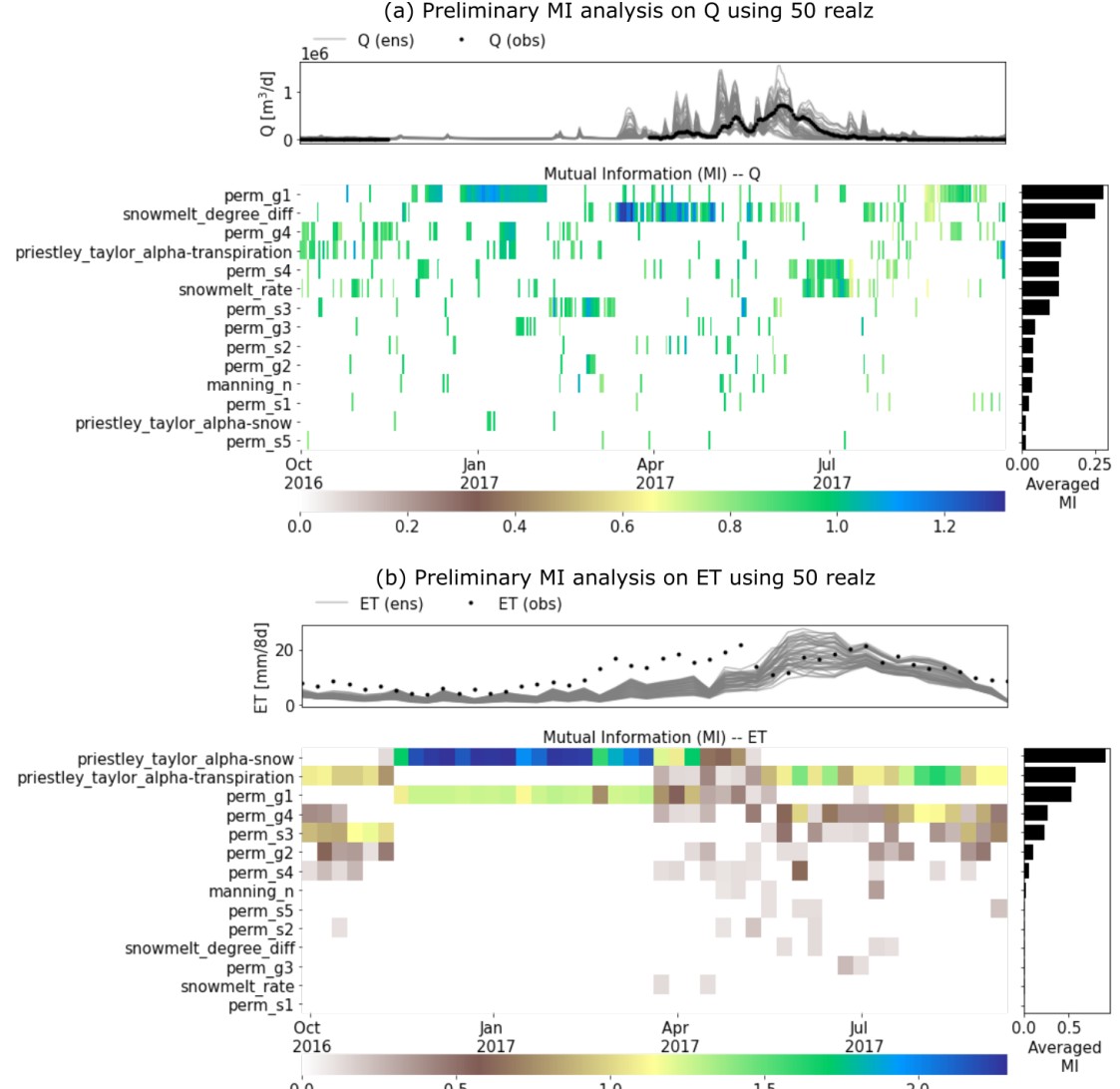

**Figure 4.** A preliminary Mutual Information (MI)-based sensitivity analysis using 50 realizations of the Advanced Terrestrial Simulator (ATS) on (a) the discharge (Q) and (b) evapotranspiration (ET). The top panel of each subplot shows the ATS ensemble (gray) and the corresponding observations (black). (Note that only water year 2016 is completed by all 50 runs).

Figure 6). The main permeability in the soil layer (i.e., perm_s3), for example, now shows higher and more temporally coherent sensitivity to Q (Figure 5(a)). On the other hand, perm_s4, which shows some sensitivity in the preliminary analysis, turns out to be insensitive to both Q and ET with almost zero MI at each time step. Since both Q and ET share little MI with perm_s4, we exclude it from the parameters to be calibrated. In the following, we leveraged this MI result to identify the sensitive responses (i.e., with non-zero MI) to be used as the inputs of the knowledge-informed DL inverse mapping.

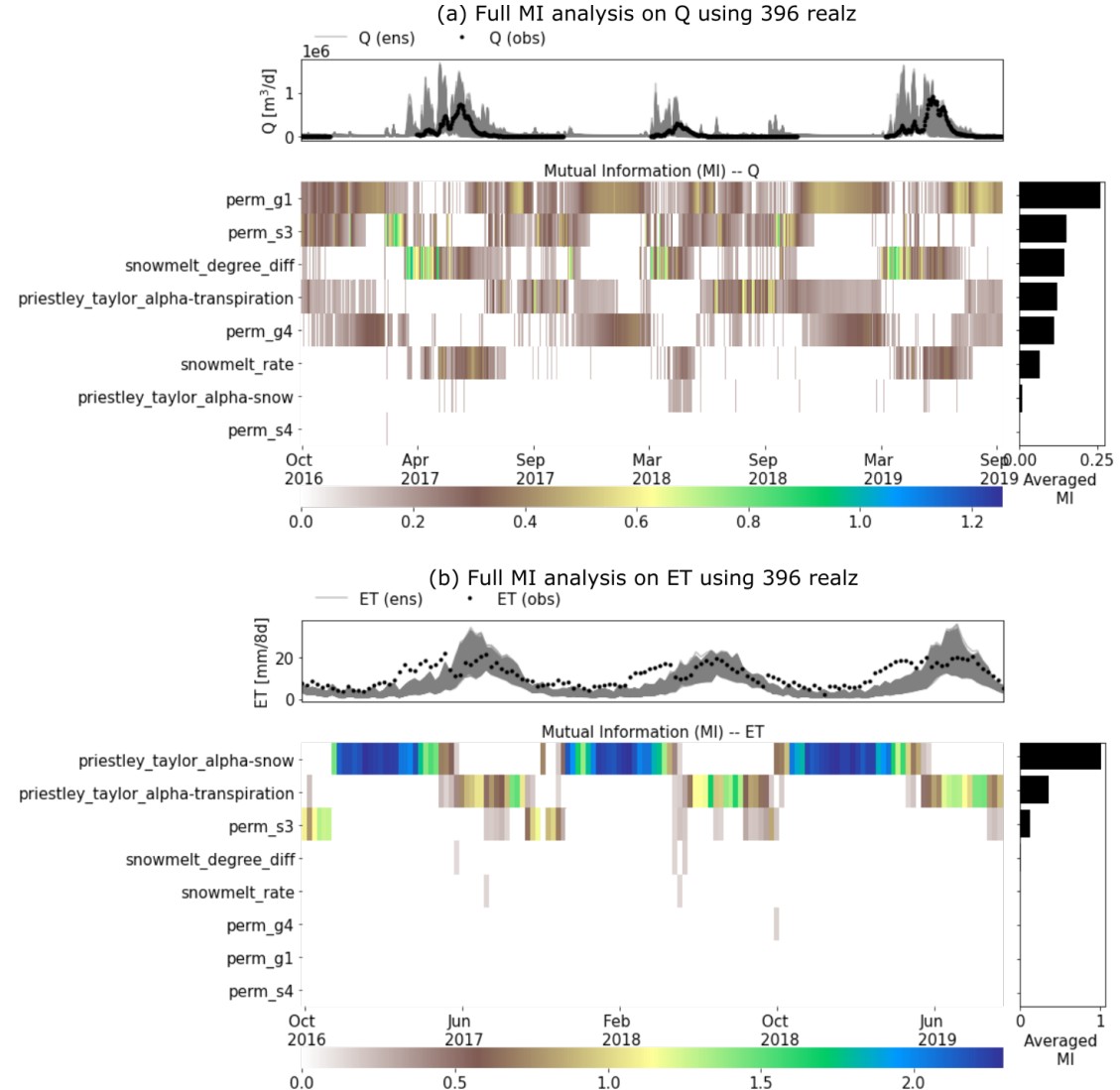

**Figure 5.** A full MI-based sensitivity analysis using 396 realizations of the Advanced Terrestrial Simulator (ATS) on (a) the discharge (Q) and (b) evapotranspiration (ET). The top panel of each subplot shows the ATS ensemble (gray) and the corresponding observations (black).

**Physical knowledge obtained by MI analysis.** The sensitivity analysis reveals the seasonal importance of these watershed characteristics to the hydrological fluxes in this area (Figure 5). During the low flow period (September through March of next year), Q is mostly controlled by the subsurface permeability (i.e., perm_g1, perm_s3, and perm_s4) which regulates both the infiltration and the groundwater movement. Transpiration also plays a role in driving the low flow dynamics through the Priestley Taylor coefficient (e.g., priestley_taylor_alpha_transpiration). During the high flow period (March through September), the snow melting process turns out to be the most critical factor in contributing to the large runoffs, which complies with the prior

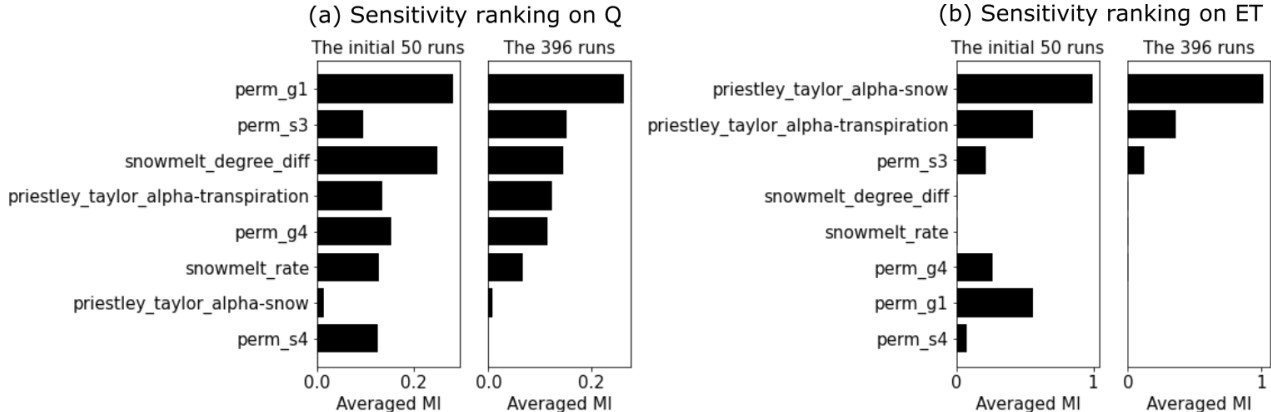

**Figure 6.** Sensitivity rankings based on mutual information (MI) on (a) the discharge (Q) and (b) evapotranspiration (ET) of both the preliminary and full MI analyses.

knowledge about the dominance of the snow process in this watershed. Likewise, the total ET is by and large attributed to
315 a variety of evaporation and transpiration. Snow evaporation is the main component of the total ET in both late autumn and winter when the snow melting rarely happens. On the other hand, in warmer and high-flow seasons, transpiration becomes the dominant contributor to the total ET. The seasonable pattern of the sensitivity of each parameter not only uncovers the hydrological process in the watershed but also serves as the basis to select the most informative model responses to estimate each model parameter.

**3.2   Performances of deep learning-based inverse mappings**

The developed inverse mappings demonstrate limited overfitting issues. Figure 7 plots the training and validation loss over epochs of the seven parameters, each of which is estimated by the knowledge-informed inverse mapping using the corresponding three years of sensitive streamflows (i.e., mi-qonly-3yrs). It can be observed from the figure that both losses quickly decrease with epochs with little discrepancies. Particularly, the parameters sharing with higher mutual information with streamflows
show faster convergences of the loss function and do not have overfitting problem (e.g., perm_s3 and snowmelt_degree_diff; see Figure 6(a)). The discrepancy between training and validation losses gets slightly larger for less sensitive parameters (e.g., perm_g4) where streamflow is less informative in parameter estimation. Indeed, informative model responses can provide better parameter estimations, thus reducing the overfitting impact. The limited impact of overfitting is also evident from the NSE and mKGE barplots of the training, validation, and test sets of all the inverse mappings (see Figures A3 and A4), where most
mappings have similar performances on parameter estimations among the three sets.

   **The improved parameter estimation using knowledge-informed inverse mapping.** We further compare the performances of the inverse mappings with and without being knowledge informed, represented by blue and green colors, respectively, in Figure 8. Blank, cross, and circle textures stand for the mapping taking as inputs only discharge (qonly), only evapotranspiration (etonly), and both (qet), respectively. Overall, most inverse mappings are trained well with both mKGE and NSE greater than

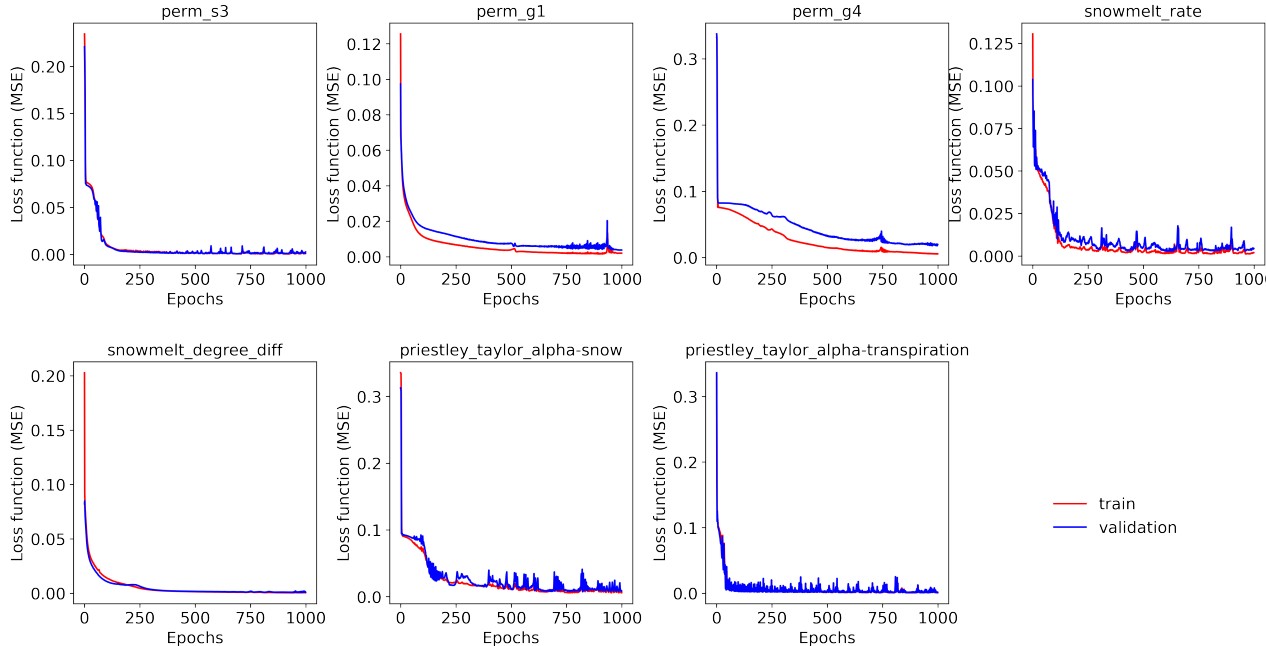

**Figure 7.** The mean squares error (MSE) loss functions of the training and validation data over 1000 epochs for the sever parameters, each of which is estimated by a knowledge-guided inverse mapping (mi-qonly-3yrs) by using the corresponding sensitive streamflow during the model calibration period (i.e., water years 2017 through 2019).

0.8, with a few exceptions (e.g., the two snow melting coefficients, perm_g1, and perm_g4 using etonly-3yr and mi-etonly-3yr). The inferior estimations of the four parameters are due to their minimal MI shared with ET (Figure 5) such that ET dynamics are insufficient to inform the two parameters.

      It can be observed from Figure 8 that a knowledge-informed inverse mapping (i.e., the blueish/black bars) generally outperforms and has higher mKGE/NSE than its counterpart that uses all the observed time steps as inputs (i.e., the greenish

bars). Noticeably, the NSE of knowledge-informed mapping increases when estimating the two Priestley-Taylor coefficients, the two snow melting parameters (except that estimated by mi-etonly-3yrs), and perm_s3, regardless of which model response is used in calibration. The extent of how sensitive the two responses are to a parameter also plays a role in the performance of the inverse mappings. The two Priestley-Taylor coefficients, which are the two most sensitive parameters to ET, are better estimated by using ET than using only Q dynamics. On the other hand, using Q yields superior performance over using ET in

estimating permeability and the snow melting coefficients. As a result, when both Q and ET are used, the knowledge-guided inverse mappings (the black bars) turn out to be the best calibration tool for most of the parameters.

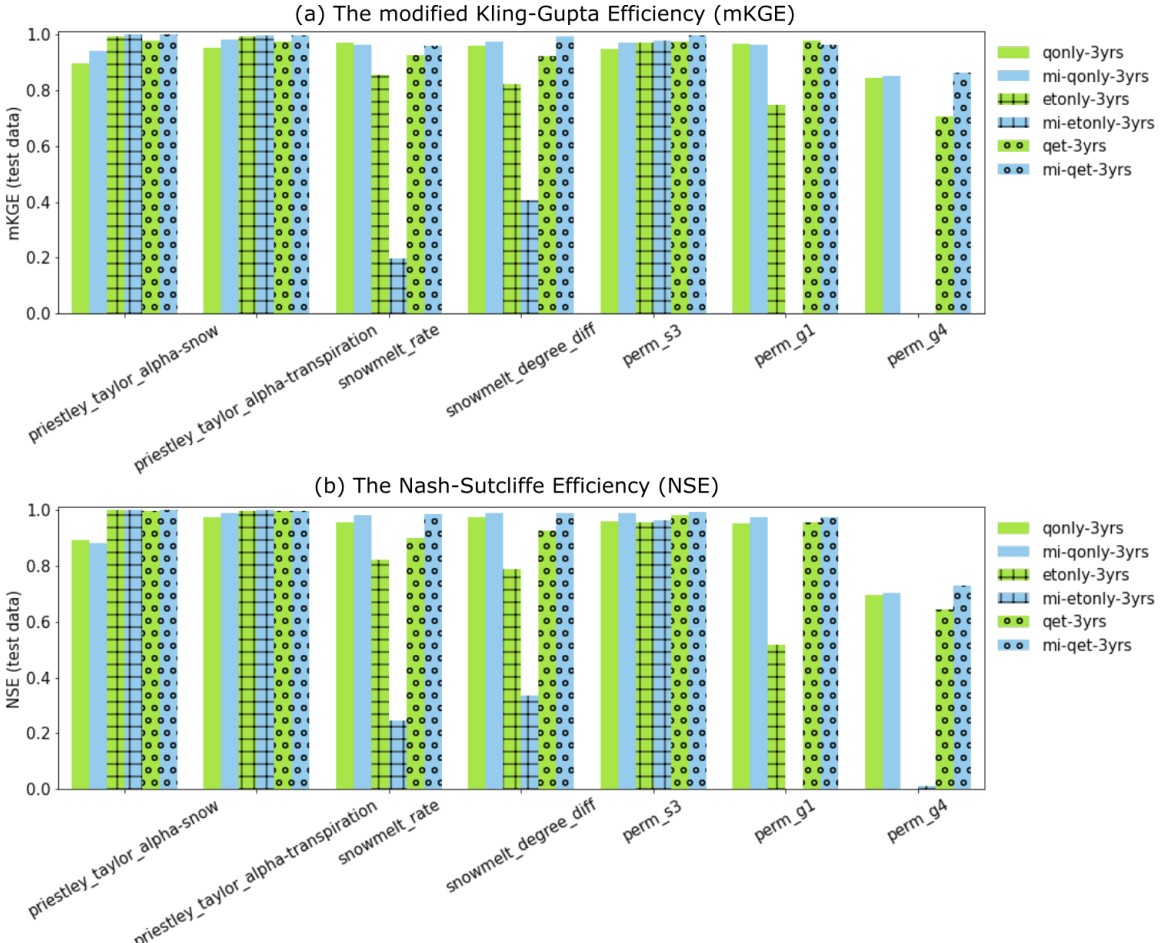

**Figure 8.** Parameter estimation performance of the developed deep learning (DL) inverse mappings on the test dataset using the model responses in the calibration period with regards to (a) the modified Kling-Gupta Efficiency (mKGE) and (b) the Nash-Sutcliffe Efficiency (NSE). Green and light blue represent the mappings without and with being knowledge informed, respectively. Blank, cross, and circle textures are used to represent the mapping using discharge only (qonly), evapotranspiration only (etonly), and both (qet), respectively.

## 3.3 Forward runs of calibrated ATS using 3yr Q/ET/Q-ET

We estimated the parameters using each of the inverse mappings (see Table A2) and performed the corresponding ATS forward runs. Figure 9 shows the Q and ET observations (the black lines) as well as the calibrated simulations during calibration (the blue lines) and evaluation (the cyan lines) periods, with their 1-to-1 scatter plots shown on the side. Overall, the parameter estimated by using knowledge-informed inverse mapping improves the calibrated Q simulation. When using only Q for calibration, mKGE increases from 0.65 (qonly-3yrs) to 0.80 (mi-qonly-3yrs) in the model calibration period.

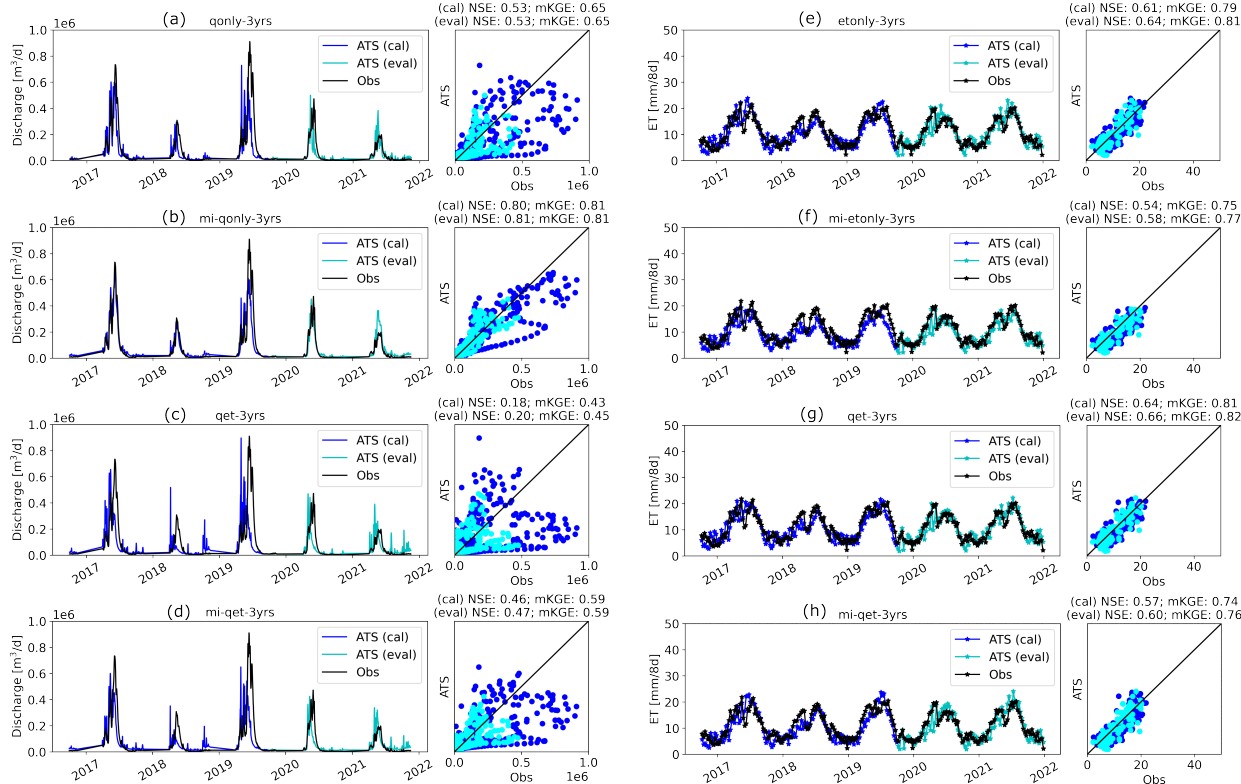

**Figure 9.** Forward runs of the Advanced Terrestrial Simulator (ATS) based on the estimated parameters from inverse mappings using the observed responses of the model calibration period in Table 1. (a-d) are the simulated discharge (Q) using qonly-3yrs, mi-qonly-3yrs, qet-3yrs, and mi-qet-3yrs, respectively. (e-h) are the simulated evapotranspiration (ET) using etonly-3yrs, mi-etonly-3yrs, qet-3yrs, and mi-qet-3yrs, respectively. (Blue and cyan colors represent the ATS forward runs at the model calibration and evaluation period, separately.)

**Adaptability of the calibrated model in the evaluation period.** For both Q and ET, both NSE and mKGE of the evaluation period (the cyan lines) are astonishingly close, if not identical, to that of the calibration period (the blue lines). Whenever the
calibrated model shows improvement using the knowledge-informed inverse mapping (such as the comparison between qonly-3yrs and mi-qonly-3yrs), we can observe the corresponding improvement in the evaluation period. Such consistent performance between the two periods suggests the robustness of the estimated parameters to climate sensitivity.

**The extrapolation issue of ET observations.** While using knowledge-informed inverse mapping improves the calibrated Q, ET simulations deteriorate, with NSE decreasing from ∼0.6 (etonly-3yrs) to ∼0.5 (mi-etonly-3yrs). This surprising result is
probably attributed to both the extrapolation issue of ET observations and the high uncertainty of the remote sensing product. Compared with the ensemble simulation of Q (Figure 5(a)) that captures most observed Q, a majority of ET observations exceed the range of the ATS ensemble of ET during the low ET period each year (i.e., wet seasons or September through May next year; see Figure 5(b)). While it is possible that the defined sampling ranges of the two Priestley Taylor coefficients in

Table A1 are too limited to provide sufficient variations of ET dynamics, the uncertainty of the MODIS ET product also plays a role here (Khan et al., 2018; Xu et al., 2019). Xu et al. (2019) show that the MODIS ET product has much poorer performance and higher uncertainty in the Colorado Basin than in most of the remaining areas in the United States. The large uncertainty of this remote sensing product probably results from the increasing error in the satellite data caused by the cloudier sky in the mountainous region (Senay et al., 2013), particularly during the dry seasons (i.e., May through September) (Xu et al., 2019). In other words, although the ET ensemble gives a better coverage on the observations in the dry seasons than the wet seasons (Figure 5(b)), that could be due to the underestimation of the MODIS ET in the dry period with high ET such that the mismatch between the ET ensemble and the observed ET could be probably more significant.

One consequence of these ET 'outliers' is the inability of the inverse mapping to reasonably estimate the parameters. The resulting estimations of priestley_taylor_alpha-snow, whose sensitivities with ET mainly occur during the low ET period, greatly surpass its maximum sample threshold (i.e., 1.2; see Table A1) and range from 1.8 to 2.1 (see Table A2). These unreliable parameter estimations make the knowledge guidance less valid. Another evidence of the adverse impact of the extrapolation issue is the inferior simulation on Q when both Q and ET are used in calibration (e.g., qet-3yrs and mi-qet-3yrs). While using knowledge-informed inverse mapping still increases the NSE to 0.47 (mi-qet-3yrs) from 0.19 (qet-3yrs), both are worse than that of calibrating against Q alone (i.e., qonly-3yrs and mi-qonly-3yrs). This extrapolation issue underscores the significance of defining the acceptable parameter sampling ranges and understanding the observation uncertainty before calibration. We thus performed the remaining analysis using only Q for calibration to avoid the ET extrapolation impact.

### 3.4 Forward runs of calibrated ATS using multi-years Q

To investigate how wet/dry water years impact the model calibration, we further employed the knowledge-informed inverse mapping to calibrate ATS on different numbers of years of Q ensemble. To this end, we used one year, two years, and all three years of Q to develop the inverse mapping for model calibration. Figure 10 plots the calibrated ATS forward runs of Q, along with the observations. First, the performances of the calibration and evaluation periods are closely identical to each other, again, illustrating the applicability of the calibrated model in an unseen time period. Second, we find that increasing the number of observed years in calibration does not necessarily improve the performance of ATS simulation on discharge. This is, in fact, consistent with the performance of the corresponding inverse mapping on the test dataset (see Figure A5 in the appendix). In other words, using only one year of observations can yield a similar calibration result with using multiple years at this watershed. This can be attributed to the similar seasonal cycle of the whole-year discharge dynamics such that multi-year dynamics do not necessarily add more information to improve the calibration. This multi-year analysis underscores the potential of using fewer years than the number of available observed years in model calibration, which can save the computational time for one ensemble run and lead to more ensemble simulations given a fixed computational budget.

**The significance of dry year dynamics.** Despite the similar calibration results between using one and multiple years, we find that by including the dry year (i.e., the 2nd year), the calibration greatly improved the simulation of Q over the scenarios using only wet years dynamics. Using only the dry year (i.e., 2nd year or mi-qonly-1yr-2) in calibration generates the best-simulated Q with NSE and mKGE above 0.8 for both calibration and evaluation periods. It outperforms the simulated Q using

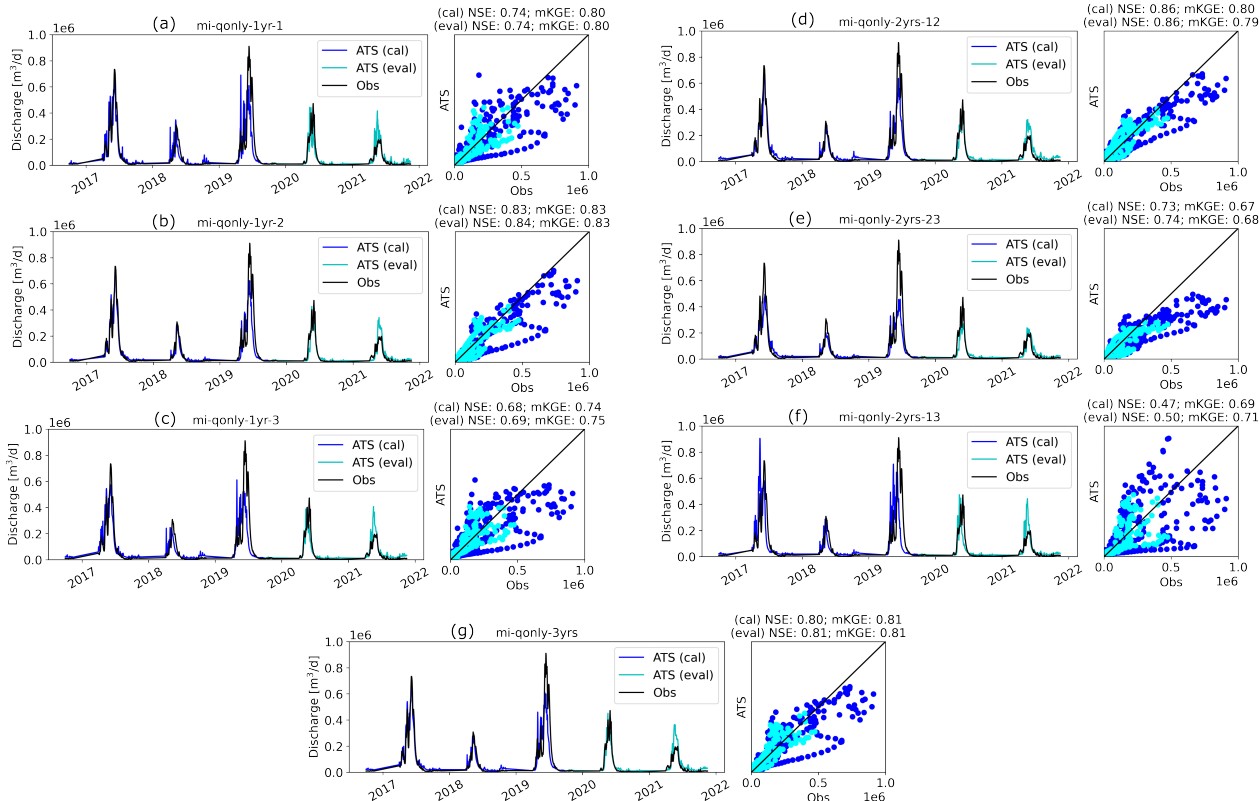

**Figure 10.** Forward runs of the Advanced Terrestrial Simulator (ATS) based on the estimated parameters from knowledge-informed inverse mappings using multiple years of observed discharge (Q) during the model calibration period in Table 1. (a-c) are the simulated discharge (Q) using each of the three water years to calibrate (i.e., mi-qonly-1yr-1, mi-qonly-1yr-2, and mi-qonly-1yr-3). (d-f) are the simulated discharge (Q) using two of the three water years to calibrate (i.e., mi-qonly-2yrs-12, mi-qonly-2yrs-23, and mi-qonly-2yrs-13). (g) is the simulated discharge using all three years to calibrate (i.e., mi-qonly-3yrs). (Blue and cyan colors represent the ATS forward runs at the model calibration and evaluation period, separately.)

either of the other two wet years (i.e., mi-qonly-1yr-1 and mi-qonly-1yr-3) with the two metrics varying from 0.68 to 0.8. Particularly, mi-qonly-1yr-3, that uses the 3rd and wettest year in calibration, generates the worst simulation of Q among the three inverse mappings using one-year dynamics. For the inverse mappings using two years, the simulated Q of the mappings including the dry year (i.e., mi-qonly-2yrs-12 and mi-qonly-2yrs-23) are better than that of using the two wet years (i.e., mi-qonly-2yrs-13). We also observe that mi-qonly-2yrs-12 outperforms mi-qonly-2yrs-23 in the simulation of Q, which probably results from the better performance of using the 1st year (mi-qonly-1yr-1) only than using the wetter 3rd year (mi-qonly-1yr-3). And using all three years (i.e., mi-qonly-3yrs) also guarantees reasonably well simulations of Q with both metrics equal to or slightly above 0.8.

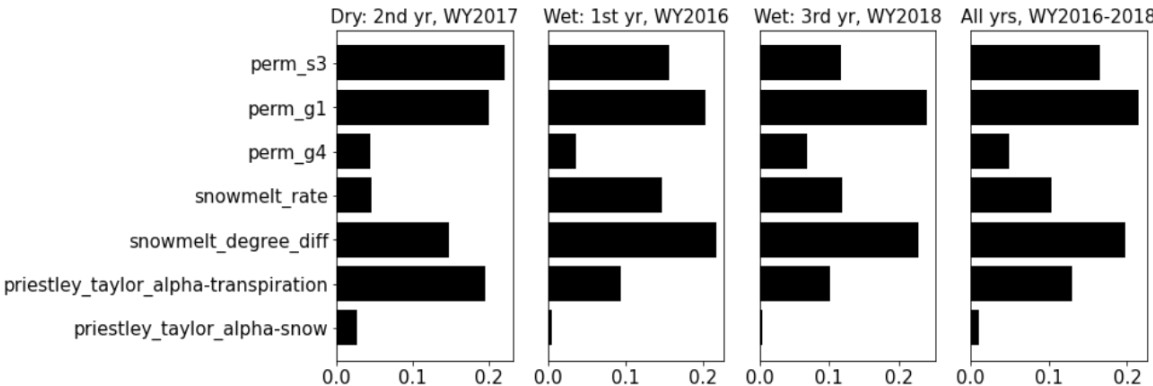

**Figure 11.** Yearly averaged Mutual Information (MI) between the parameters and discharge (Q) computed from the full MI analysis result in Figure 5(a). Left to right: the dry year (i.e., water year (WY) 2018), the first wet year (i.e., WY2017), the second wet year (i.e., WY2019), and all the three years.

The enhanced calibration performance by including the dry year dynamics is probably due to the improved dependencies between discharge and subsurface features during low flow periods. This can be observed in the yearly averaged MI bar plots in Figure 11. With reduced sensitivities on the snow parameters, the dry year shows that the dominant permeability in the soil layer (i.e., perm_s3) proves to be the most sensitive parameter to discharge (with averaged MI 0.2). Meanwhile, the averaged MI of perm_s3 decreases to only around 0.1 in the 3rd year by using which the calibrated Q is the worst with NSE=0.67 and mKGE=0.73. In fact, all the inverse mappings using the dry year consistently estimates higher perm_s3 (i.e., 10.9 $\log_{10}(m^2)$) than the other estimates ranging from -11.4 to -12.2 $\log_{10}(m^2)$ (see Table A3 in the appendix). The higher soil permeability estimated by the dry year has a better capability in draining the surface water and thus reduces the outlet discharge spikes during late spring and early autumn, thus yielding the simulations of Q more consistent with the observations plotted in Figure 10.

Our finding on the significance of dry year discharge in model calibration indirectly supports some recent studies. Pool et al. (2019) found that high flow provides limited information to calibrate models in snow-dominated catchments. This is mainly because there are fewer discharge fluctuations during snow melting or high flow period than rainfall-fed catchments (Viviroli and Seibert, 2015). The decreased role of high flow, in turn, enhances the importance of the low flow period in calibration, particularly in dry years. Indeed, in this watershed, we do observe stronger diurnal discharge fluctuations during the low flow period of the dry year (i.e., WY2018) than the other two wetter years (see Figure A6 in the appendix), which facilitates the better calibration result using observations from the dry year.

**Impact of observation errors.** For each of the seven inverse mappings, we set 5% observational error and generated 100 sets of estimated parameters by using 100 realizations of noisy observed discharges. Figure 12 shows the barplots of these ensemble estimations of parameters. Overall, most estimated parameters (i.e., priestley_taylor_alpha-transpiration, snowmelt_degree_diff, perm_s3, and perm_g1) show little variability, indicating the robustness of the trained inverse mappings. Although there is higher variability for the other parameters (i.e., priestley_taylor_alpha-snow, snowmelt_rate, and

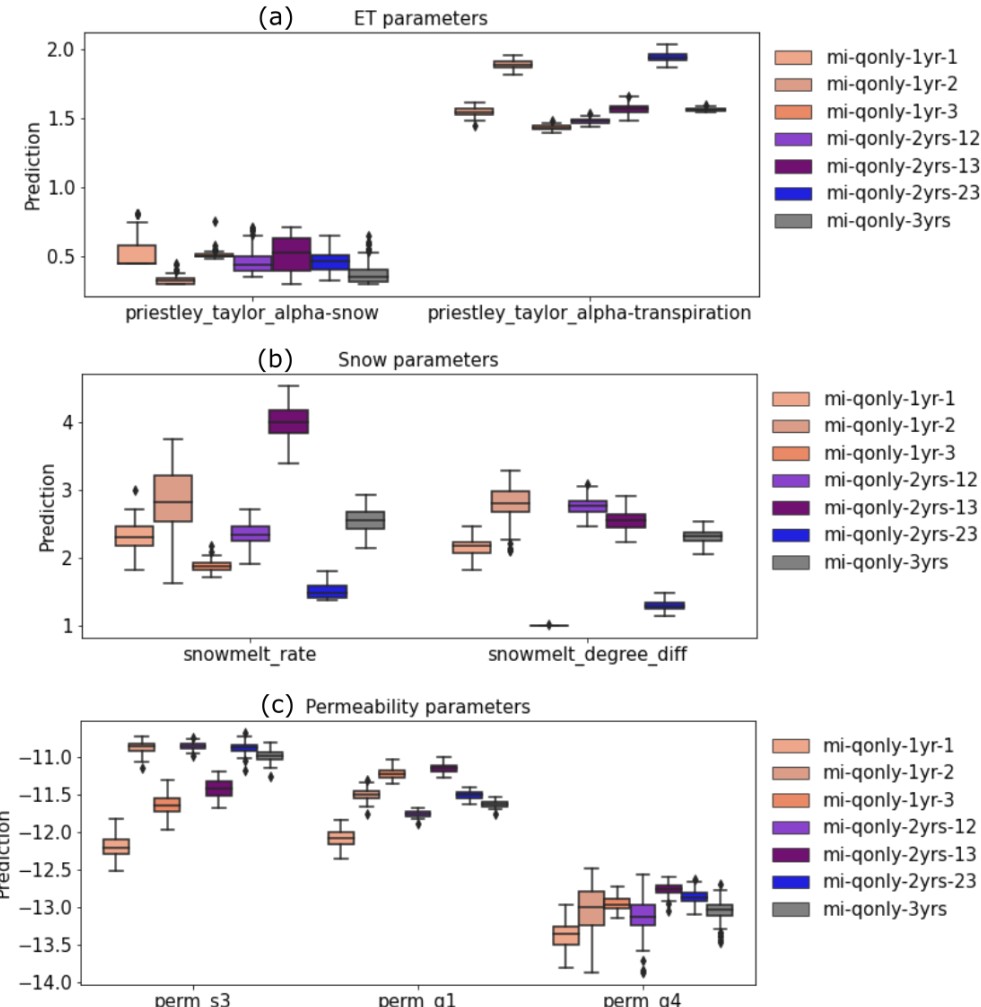

**Figure 12.** Box plots of ensemble parameters estimated knowledge-informed inverse mappings of multiple years of discharge (Q) using 100 realizations of noisy discharge observations with 5% observation error, categorized into (a) evapotranspiration (ET) parameters; (b) snow parameters; and (c) permeability parameters.

perm_g4), these parameters share less MI with Q than the others (Figure 11). Therefore, the larger variations of these less sensitive parameters have little impact on the corresponding ATS forward runs. Indeed, as shown in Figure 13, the ATS ensemble runs using inverse mapping mi-qonly-3yrs demonstrate little variations of the ensemble runs of the discharges compared with the corresponding run without observational error, which verifies the robustness of these inverse mappings against the observational error.

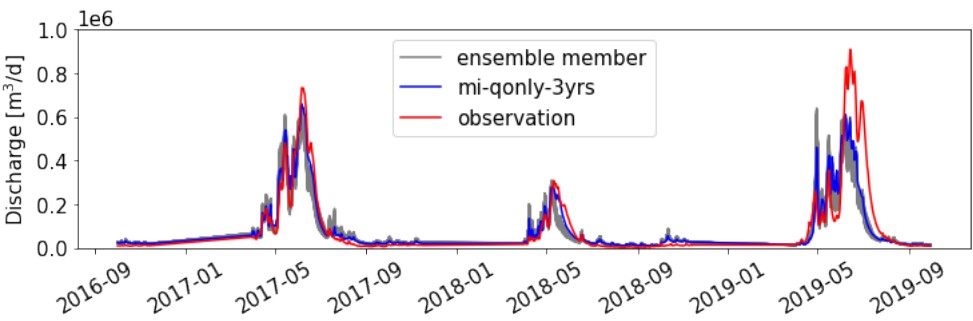

**Figure 13.** Ensemble forward runs of the Advanced Terrestrial Simulator (ATS) on discharge using 100 ensemble parameters estimated by mi-qonly-3yr through noisy discharge observations with 5% observation error.

## 4   Conclusions

We developed a novel model calibration methodology that leverages MI-based sensitivity analysis to guide the architecture of the DL-assisted inverse mapping based on only a few hundred realizations. A case study in the Coal Creek watershed shows that the calibrated ATS using such knowledge-informed DL simulates the discharge well with both mKGE and NSE up to 0.8, outperforming its counterpart that does not adopt the knowledge guidance and only achieves NSE and mKGE around 0.5∼0.6 (Figures 9(a) and (b)).

The proposed hierarchical way of sensitivity analysis efficiently utilizes the available limited computational resource through a combination of a prescreening analysis and then a full analysis. Although the prescreening using 50 model runs does not theoretically exclude a false negative case that a sensitive parameter is classified as insensitive, the statistical significance test is able to improve the estimation of mutual information in Figure 4 thus facilitating narrowing down an "accurate" list of parameters to be estimated. Based on the shortened parameter list, a full sensitivity analysis is successfully performed using nearly 400 model runs and provides physically meaningful results on the dependency between the parameters and model responses in Figure 5.

Despite the improved streamflow simulations, we observe the adverse impact of the observation outliers when calibrating ATS against remote sensing ET product (Figures 9(e)-(h)). These outliers deteriorate the performance of the inverse mapping in parameter estimation, thereby worsening the calibrated ATS forward runs. Based on the inverse mappings taking ET as inputs, the estimated priestley_taylor_alpha-snow, which is the most sensitive to ET (Figure 5(b)), greatly exceeds the range of the parameter sample range (Tables A1 and A2). While the uncertainty of MODIS ET (Khan et al., 2018; Xu et al., 2019) also contributes to this extrapolation issue, this result underscores the significance of suitably defining the parameter sample range to assure the uncertainty of ensemble simulations covers the observations.

We further find that using one or two years of observations in calibration yields similar or even better results than that of using three years. This encouraging result highlights the importance of including abnormal year data for model calibration. This would significantly reduce the computational cost of each model run, increasing the number of model ensemble realizations

used in developing calibration techniques. While earlier studies found that several years of discharge observations are necessary to achieve a reasonable model calibration (Sorooshian et al., 1983; Yapo et al., 1996; Perrin et al., 2007) by using either a semi-distributed or a bucket model, we suspect that employing a fully integrated and high resolution model like ATS can greatly reduce the errors due to the improved physical representation in the model, thus requiring longer observation periods for calibration. Hence, future work can focus on developing a systematical approach in identifying the most 'important' observed

period in model calibration.

The recently emerged knowledge-guided DL is swiftly gaining popularity in earth science. This study demonstrates one of its applications in calibrating a computationally expensive hydrological model. The developed methodology can be readily adopted in other watersheds to calibrate a different model. One potential future work is to develop a unified inverse modeling framework for multiple basins, where the atmospheric forcings and basin characteristics can be also used as the inputs of

465 the inverse mappings in addition to the realization-dependent model responses. With the increasing complexity of earth system models, we believe such knowledge-guided DL calibration can pave the way for efficient yet effective model calibration without increasing significant computational demand.

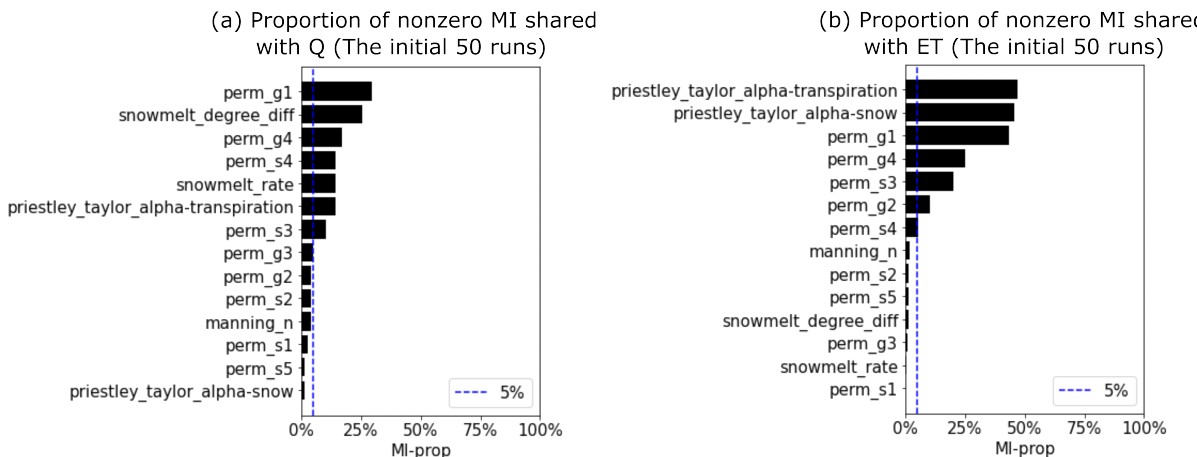

**Figure A1.** Proportion of nonzero mutual information (MI) between the parameters and the simulated streamflow Q (a) and evapotranspiration ET (b) based on the initial 50 ensemble runs. (Blue dashed line is the threshold (i.e., 5%) for selecting sensitive parameters for a full mutual information using 396 realizations.)

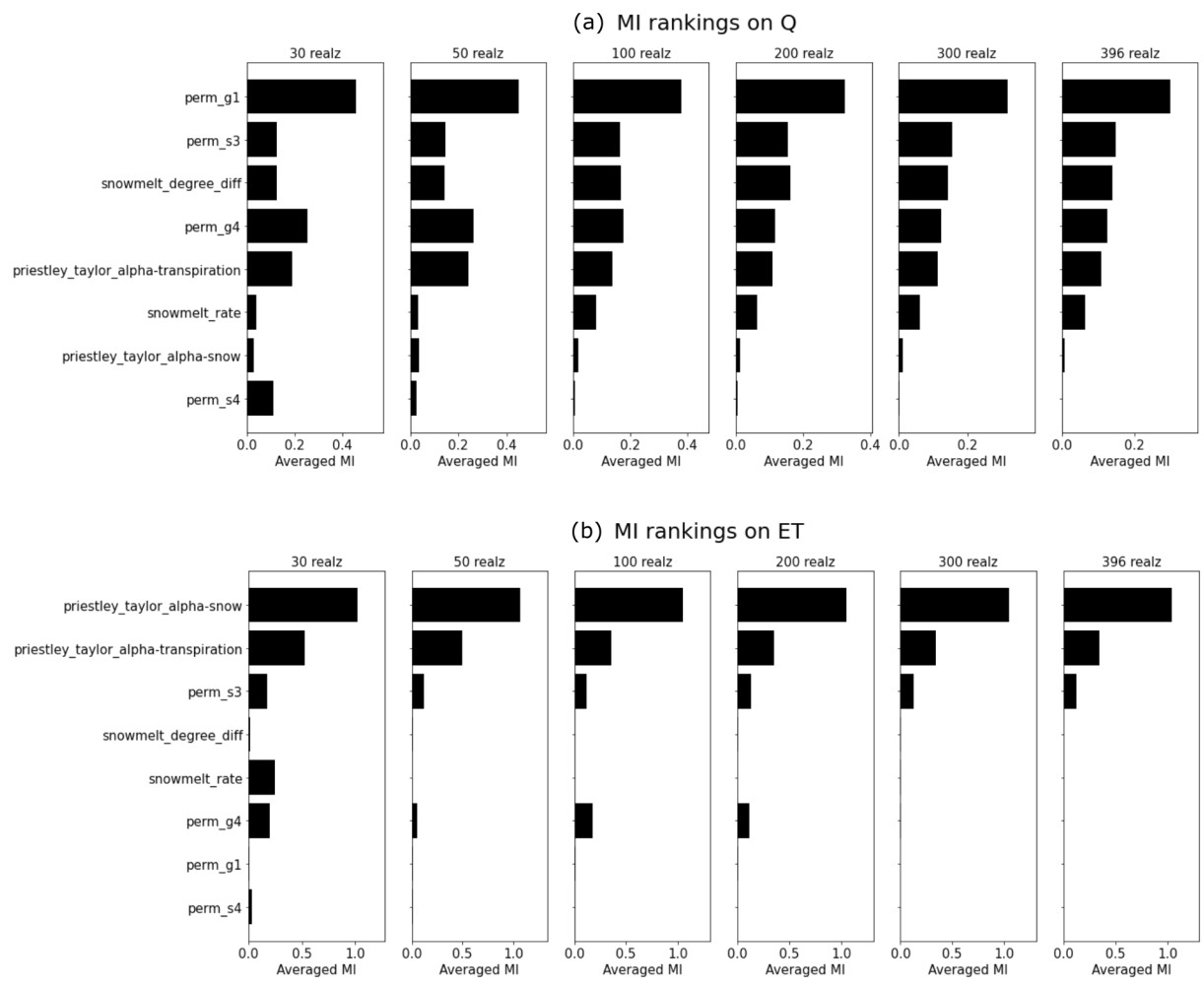

**Figure A2.** Convergence of the parameter sensitivity rankings by using the averaged mutual information (MI) on discharge (Q) and evapo-transpiration (ET) using 30, 50, 100, 200, 300, and 396 model realizations.

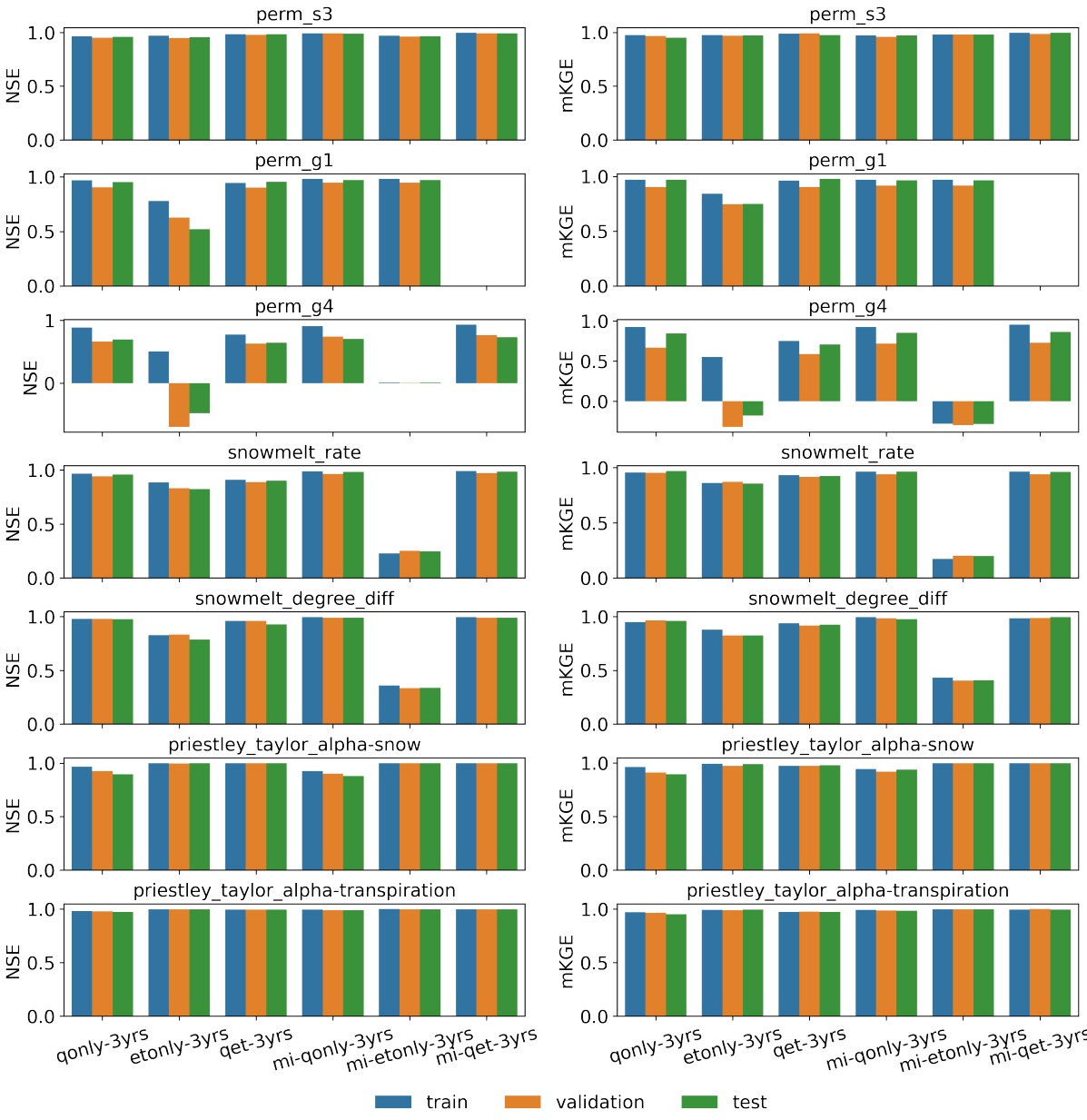

**Figure A3.** Parameter estimation performance of the developed deep learning (DL) inverse mapping on the train, validation, and test datasets using three-years of discharges (Q) and evatranspiration (ET) with regards to (right) the modified Kling-Gupta Efficiency (mKGE) and (left) the Nash-Sutcliffe Efficiency (NSE).

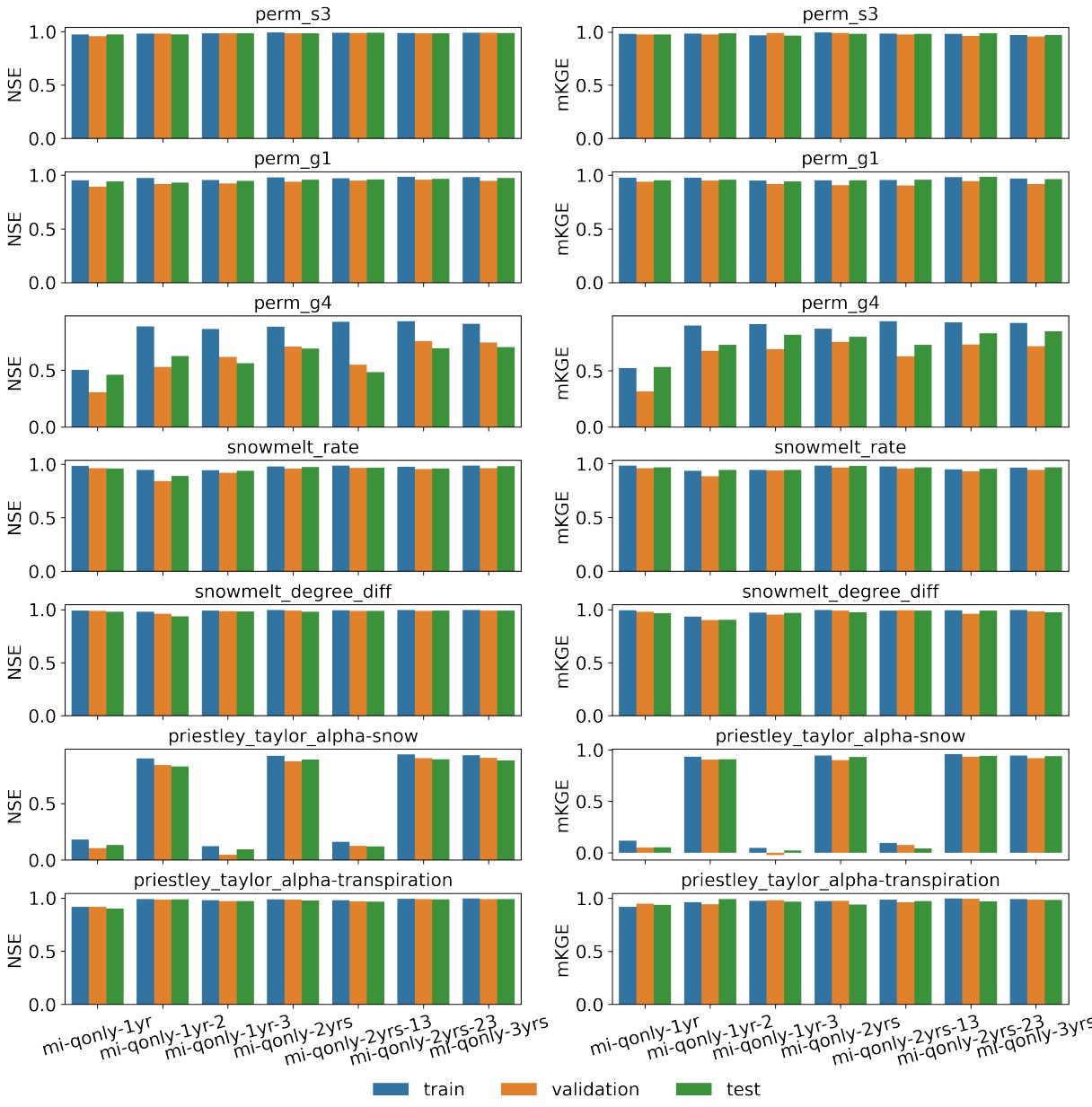

**Figure A4.** Parameter estimation performance of the developed deep learning (DL) inverse mapping on the train, validation, and test datasets using multi-years of discharges (Q) with regards to (right) the modified Kling-Gupta Efficiency (mKGE) and (left) the Nash-Sutcliffe Efficiency (NSE).

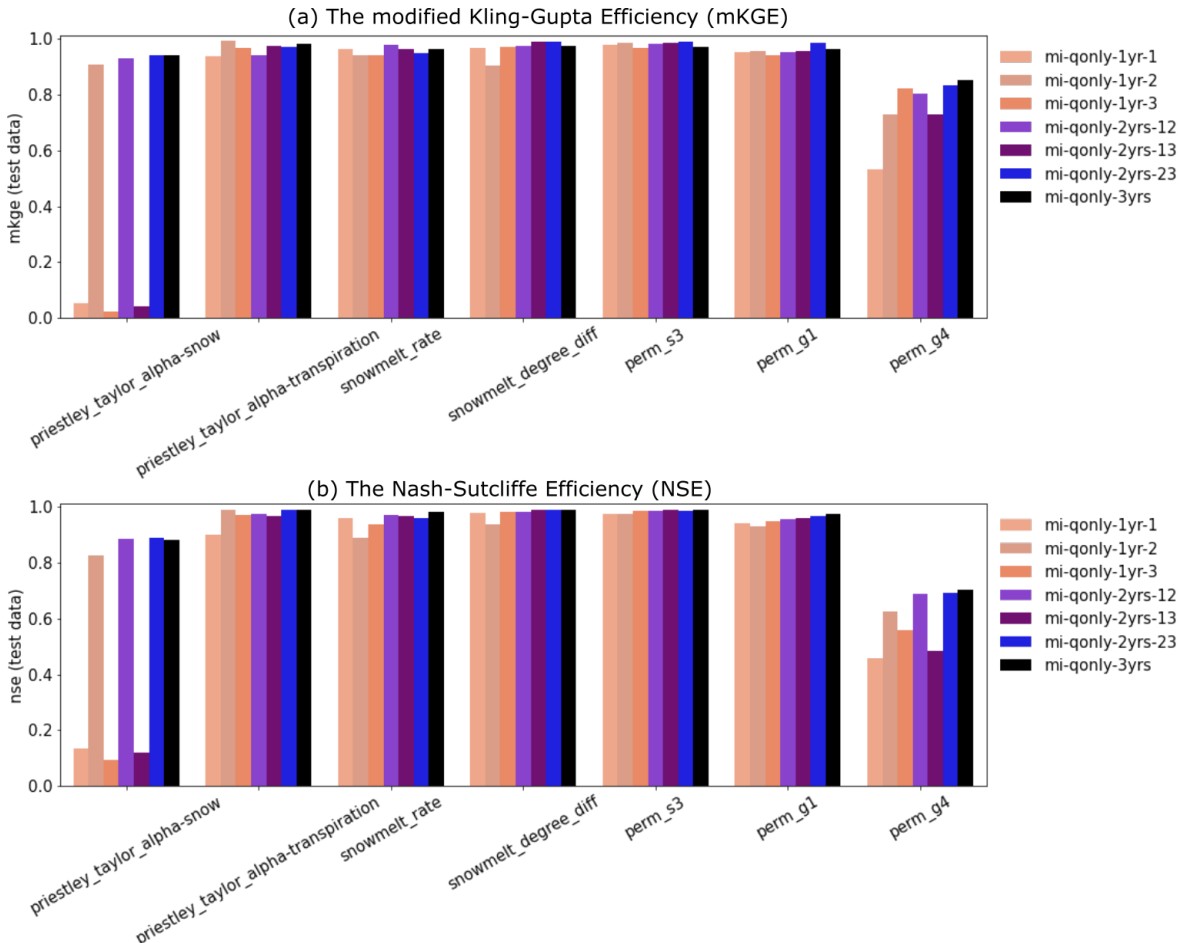

**Figure A5.** Performance of the developed deep learning (DL) inverse mapping on the test dataset using multi-years discharges (Q) with regards to (a) the modified Kling-Gupta Efficiency (mKGE) and (b) the Nash-Sutcliffe Efficiency (NSE).

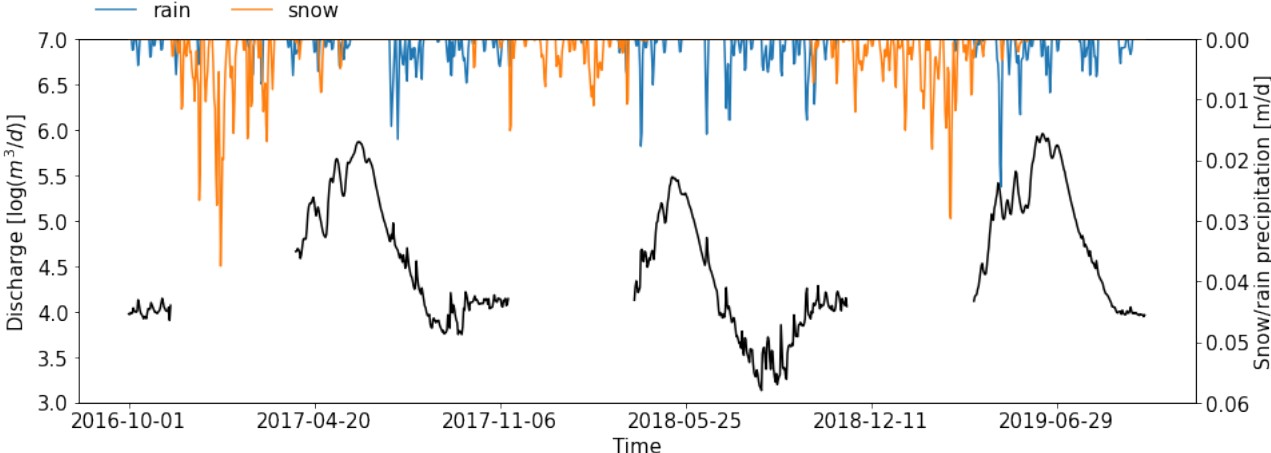

**Figure A6.** Logarithmic discharge observation from water year 2017 through water year 2019 at the USGS gage station 09111250. The top panel shows the corresponding snow (yellow) and rain (blue) precipitation.

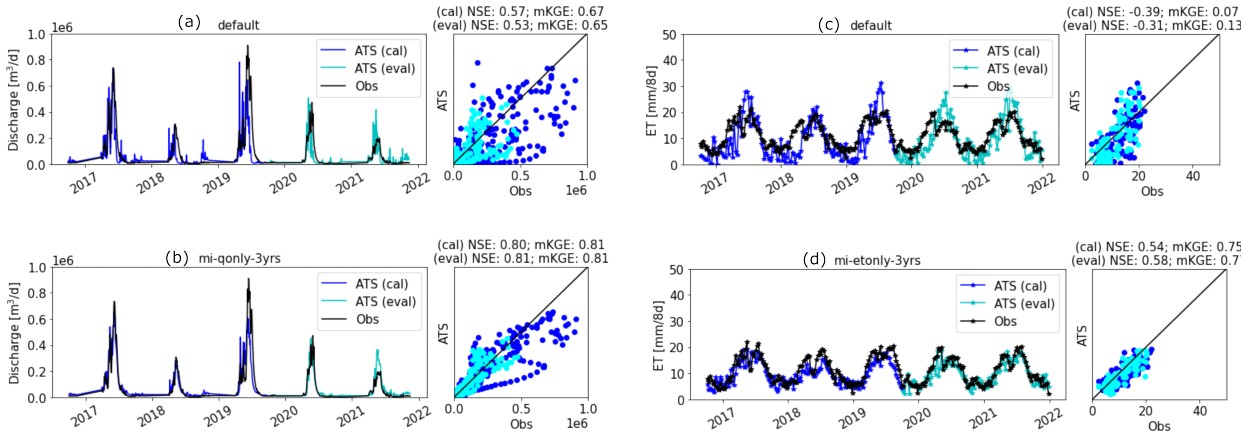

**Figure A7.** Calibrated ATS forward runs versus the default forward runs during the calibration period regarding the discharge ((a) and (b)) and evapotranspiration ((c) and (d)). (The default run uses the following parameters: priestley_taylor_alpha-snow=1.2, priestley_taylor_alpha-transpiration=1.2, snowmelt_rate=2.74, snowmelt_degree_diff=3.0, manning_n=0.15, perm_s1=2.83e-12, perm_s2=9.5e-13, perm_s3=1.95e-12, perm_s4=4.74e-12, perm_s5=2.05e-13, perm_g1=3.16e-13, perm_g2=3.02e-11, perm_g3=6.31e-16, perm_g4=1e-13.)

Table A1: List of the parameters used in the preliminary or full mutual information analysis.

| Parameter name | Description [unit] | Min | Max | Used in Full MI analysis? |
|---|---|---|---|---|
| perm_s1 | permeability of the soil layer s1 $[\log m^2]$ | -12.548 | -10.548 | No |
| perm_s2 | permeability of the soil layer s2 $[\log m^2]$ | -13.022 | -11.022 | No |
| perm_s3 | permeability of the soil layer s3 $[\log m^2]$ | -12.710 | -10.710 | Yes |
| perm_s4 | permeability of the soil layer s4 $[\log m^2]$ | -12.324 | -10.324 | Yes |
| perm_s5 | permeability of the soil layer s5 $[\log m^2]$ | -13.688 | -11.688 | No |
| perm_g1 | permeability of the geological layer g1 $[\log m^2]$ | -13.500 | -11.500 | Yes |
| perm_g2 | permeability of the geological layer g2 $[\log m^2]$ | -11.520 | -9.520 | No |
| perm_g3 | permeability of the geological layer g3 $[\log m^2]$ | -16.200 | -14.200 | No |
| perm_g4 | permeability of the geological layer g4 $[\log m^2]$ | -14.000 | -12.000 | Yes |
| snowmelt_rate | snow melt rate $[mm\ day^{-1}\ C^{-1}]$ | 1.37 | 5.48 | Yes |
| snowmelt_degree_diff | air-snow temperature difference [deg C] | 1 | 5 | Yes |
| manning_n | Manning's coefficient [-] | 0.02 | 0.2 | No |
| priestley_taylor_alpha-snow | Priestley Taylor coefficient of canopy transpiration [-] | 0.3 | 1.2 | Yes |
| priestley_taylor_alpha-transpiration | Priestley Taylor coefficient of snow evaporation [-] | 0.3 | 1.2 | Yes |

Table A2: Esitmated parameters from the inverse mappings using three years of Q/ET/Q and ET listed in Table 1 (Note that perm_g1 is not estimated by mi-etonly-3yrs because ET is not sensitive to perm_g1 at the analyzed three water years shown in Figure 5).

| | Inverse mapping | | | | | |
|---|---|---|---|---|---|---|
| | qonly-3yrs | etonly-3yrs | qet-3yrs | mi-qonly-3yrs | mi-etonly-3yrs | mi-qet-3yrs |
| perm_s3 | -11.859 | -12.720 | -12.715 | -10.992 | -12.709 | -12.708 |
| perm_g1 | -11.790 | -10.906 | -10.722 | -10.451 | n/a | -10.451 |
| perm_g4 | -12.603 | -10.150 | -13.514 | -12.536 | -12.791 | -12.276 |
| snowmelt_rate | 2.538 | 4.289 | 2.880 | 2.569 | 2.660 | 2.852 |
| snowmelt_degree_diff | 1.697 | 7.755 | 1.822 | 2.324 | 2.422 | 2.365 |
| priestley_taylor_alpha-snow | 0.296 | 2.130 | 2.135 | 0.354 | 1.925 | 1.834 |
| priestley_taylor_alpha-transpiration | 2.018 | 0.665 | 0.746 | 1.564 | 0.515 | 0.846 |

Table A3: Esitmated parameters from the inverse mappings using multi-years of Q listed in Table 1.

| | Knowledge-informed inverse mappings only using discharge (i.e., mi-qonly-*) | | | | | | |
| --- | --- | --- | --- | --- | --- | --- | --- |
| | 1yr-1 | 1yr-2 | 1yr-3 | 2yr-12 | 2yr-13 | 2yr-23 | 3yrs |
| perm_s3 | -12.185 | -10.889 | -11.629 | -10.869 | -11.410 | -10.881 | -10.992 |
| perm_g1 | -10.917 | -10.308 | -10.050 | -10.580 | -9.978 | -10.333 | -10.451 |
| perm_g4 | -12.893 | -12.522 | -12.498 | -12.637 | -12.280 | -12.363 | -12.536 |
| snowmelt_rate | 2.308 | 2.856 | 1.879 | 2.365 | 4.003 | 1.523 | 2.569 |
| snowmelt_degree_diff | 2.149 | 2.818 | 1.013 | 2.767 | 2.543 | 1.290 | 2.324 |
| priestley_taylor_alpha-snow | 0.450 | 0.318 | 0.505 | 0.434 | 0.528 | 0.447 | 0.354 |
| priestley_taylor_alpha-transpiration | 1.541 | 1.883 | 1.433 | 1.478 | 1.566 | 1.942 | 1.564 |

*Author contributions.* PJ, PS, and XC designed the numerical experiment. PJ developed and trained the deep learning-assisted inverse mapping. PS performed the ATS simulations. PJ prepared the manuscripts with contributions from all co-authors. All authors provided critical feedback and inputs to the manuscript.

*Competing interests.* The contact author has declared that neither they nor their co-authors have any competing interests.

*Disclaimer.* Copernicus Publications remains neutral with regard to jurisdictional claims in published maps and institutional affiliations.

*Code and data availability.* The data and scripts of this work are available at: https://gitlab.pnnl.gov/sbrsfa/exasheds/knowledge-informed-dl-calibration.

*Acknowledgements.* This work was funded by the ExaSheds project, which was supported by the United States Department of Energy, Office of Science, Office of Biological and Environmental Research, Earth and Environmental Systems Sciences Division, Data Management Program, under Award Number DE-AC02-05CH11231. This research used resources of the National Energy Research Scientific Computing Center (NERSC), a DOE Office of Science User Facility supported by the Office of Science of the United States Department of Energy under contract DE-AC02-05CH11231. Pacific Northwest National Laboratory is operated for the DOE by Battelle Memorial Institute under contract DE-AC05-76RL01830. This paper describes objective technical results and analysis. Any subjective views or opinions that might be expressed in the paper do not necessarily represent the views of the United States Department of Energy or the United States Government. The United States Government retains and the publisher, by accepting the article for publication, acknowledges that the United States Government retains a non-exclusive, paidup, irrevocable, world-wide license to publish, or reproduce the published form of this manuscript,

or allow others to do so, for United States Government purposes. The Department of Energy will provide public access to these results

of federally sponsored research in accordance with the DOE Public Access Plan (http://energy.gov/downloads/doe-public-access-plan, last access: 20 May 2022).

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
