# Peer review of "Knowledge-Informed Deep Learning for Hydrological Model Calibration: An Application to Coal Creek Watershed in Colorado"

_Hydrology and Earth System Sciences, 2022_

## Referee Comment (RC3)

[referee-annotated manuscript omitted]

---

## Author Comment (AC1)

**Responses to the Comments of Reviewer #1 on ⟨hess-2022-282⟩**

Peishi Jiang        Pin Shuai        Alexander Sun        Maruti K. Mudunuru

Xingyuan Chen

December 28, 2022

> General comments:
>     This study showcases a deep learning optimization method for a high-resolution
> hydrologic model supported by information theory.  I appreciate the honest
> evaluation of the methodology, in-depth reasoning of the deteriorating model
> performance for ET, and examination of results and conclusions aligned with
> earlier studies.  In general, this paper is well-written with a novel contribution.
> However, I think the paper would be stronger if the authors can address the
> following comments.

Thank you for the thorough review of our manuscript. We addressed each comment as shown below.

> Model validation for climate sensitivity:  Currently, the model validation period
> overlaps with the period for calibrating ATS parameters.  I am curious whether
> the optimized parameters would be able to capture the climate sensitivity on flow
> and ET, i.e., improving the flow/ET performance outside of the calibrating period
> (2016-2019).  It would strongly support this tool's eligibility in climate change
> studies.

We now extend the simulation period to 31-12-2021, which is the end of the available Daymet forcing (see Figure 1(b)). We split the whole period into the calibration (1-10-2016 through 30-9-2019) and the evaluation (1-10-2019 through 31-12-2021) periods, used to calibrate and evaluate the climate sensitivity of the model, separately. The result in Figures 9 and 10 shows that the performances of the calibrated ATS during the evaluation period are very close to that of the calibration period, suggesting the adaptability of the estimated parameters to an uncalibrated time period. We revised the associated results and discussion as follows:

" (L232-L239) Here, we separate the entire observations in Figure 1(b) into model calibration and evaluation periods in order to assess the adaptability of the estimated parameters to an uncalibrated period. To this end, we calibrate ATS only using the simulations during water year 2017 to water year 2019 and used the remaining observations (till 31 December 2021) for model evaluation. The ensemble runs used for sensitivity analysis and inverse modeling are performed during the calibration period. The calibrated ATS forward runs were then performed on both periods and compared against the observations in Figure 1(b). We assess the performances of the calibrated models on both periods by using two scale-independent metrics: the Nash-Sutcliffe Efficiency (NSE; [4]) and the modified Kling-Gupta Efficiency (mKGE; [3]).
    ...
    (L346-L352) **Adaptability of the calibrated model in the evaluation period.** For both Q and ET, both NSE and mKGE of the evaluation period (the cyan lines) are astonishingly close, if not identical, to that of the calibration period (the blue lines). Whenever the calibrated model shows improvement using the knowledge-informed inverse mapping (such as the comparison between qonly-3yrs and mi-qonly-3yrs), we can observe the corresponding improvement in the evaluation period. Such consistent performance between the two periods suggests the robustness of the estimated parameters to climate sensitivity. "

> ET from flux tower: In this study, the authors have demonstrated that worse ET
> performance results from poor quality of MODIS ET products. In this study region,
> is there ET data from the flux tower that could be used for implementing this
> workflow? Even though the flux tower ET data has less spatial coverage, the data
> quality can be better, which might be more useful than MODIS ET when calibrating
> hydrologic parameters.

We looked into the AmeriFlux and there is no flux tower site available in this watershed. Therefore, we are not able to perform the calibration against the site-based observations.

> Specific comments:
>     L158: Can the authors elaborate on what five soil types and four geological
> types are?

They are grouped subsurface characteristics in the soil and geological layers using k-means clustering. We add the following description for a better elaboration (L161-L163): "Each clustered soil or geological type is associated with a specific set of subsurface characteristics (such as permeability), which are assigned to the corresponding grouped grid cells. These subsurface characteristics are important in controlling flow dynamics and can be estimated from hydrological observations."

> L160: A 1000-year spin-up is extremely long. Can the authors briefly explain the
> reason for this long spin-up even if it might be explained in Shuai et al 2022?

We have revised the associated text to clearly explain the motivation for the 1000-year cold or steady-state spinup (note that the cold spin-up took less than one hour to complete on 128 CPU cores due to the faster model convergence once it reached quasi-steady-state.): " (L163-L167) To ensure that the model achieved a physically appropriate initial state, two spinups were performed sequentially, including (1) a cold spinup that ran the model for 1000 years by using constant rainfall and led to steady-state condition at the end of the simulation (e.g., converged total amount of subsurface water storage) and (2) a warm spinup that was initialized by the steady-state spinup result and performed a transient simulation for 10 years (i.e., 1 October 2004 – 1 October 2014) under the Daymet forcing. "

> L162: Could the authors briefly explain how they preselected the parameters in
> this study?
>     L208: Does the MI have to be zero? If the MI between a parameter and the
> model responses is small enough, is it possible to neglect that parameter? What
> would be a proper threshold for it?
>     L249-250: Given the narrowed list, it seems that the authors eliminated the
> parameters with small MI (not zero), which slightly contradicts the previous
> statement where only parameters with zero MI would be eliminated (L208). It
> would be helpful to clarify the threshold of MI below which the parameters will
> be eliminated.

As all three comments are associated with how we performed the preliminary sensitivity analysis using mutual information, we reply to them in one thread here. In short, the preselection is based on the mutual information (MI) computed for each parameter and each response at a given time step. For a given model response (e.g., Q), we say it is sensitive to a parameter if the proportion of non-zero MI over all the time steps is greater than a given threshold (i.e., 5% in this study). For each MI calculation, we performed a statistical significance test to determine whether the computed MI is significant and set MI to zero if the test fails. So, the MI can be zero. We enriched the description in the associated texts as below:

"

" (L190-L194) In this study, we follow a similar strategy of [1] to estimate $p$ using 10 evenly divided bins along each dimension and perform SST tests to filter out any non-significant MI value with a significance level of 95% based on 100 bootstrap samples. In other words, the computed MI is set to zero if the statistical significance test fails.

...

(L205-L209) This preliminary MI analysis would allow filtering out the parameters that show little sensitivity to the model responses, thus reducing the number of parameters to be calibrated. This filtering process is performed based on whether a parameter demonstrates sufficient sensitivity across the simulation period. In this study, we selected the parameter whose proportion of the non-zero MI is larger than 5% of the overall time steps for the following full sensitivity analysis.

...

(L285-L290) Based on the proportion of nonzero MI over all the time steps (see Figure A1 in the appendix), we find that Q is mostly sensitive to (using a threshold of 5%) perm_s3, perm_s4, perm_g1, perm_g4, snowmelt_rate, snowmelt_degree_diff, and priestley_taylor_alpha_transpiration, and ET is mostly sensitive to priestley_taylor_alpha_transpiration, priestley_taylor_alpha_snow, perm_s3, perm_g1, and perm_g4. Consequently, we narrow down the parameters to be calibrated by taking the union of the two sets of parameters that show sensitivities to either Q or ET (also highlighted in Table A1). "

> L208-210:  Interesting!  Great summary!

Thank you for the generous comment.

> L215:  When training using different combinations of years, why do the authors
> only look at Q, not ET?

We do not use ET for multi-year analysis because the extrapolation issue of the ET observations deteriorates the parameter estimations using the inverse mapping, as described in Section 3.3. In other words, a multi-year analysis including ET would be questionable and not trustworthy to evaluate the impact of dry and wet years. Therefore, we performed the multi-year analysis against only Q.

> L286-287:  Please clarify whether the extrapolation issue partially or solely
> contributes to the worse MI-informed results.

The inferior calibrated ATS runs using knowledge-informed deep learning are attributed to both the extrapolation issue of the observations and the potential high uncertainty of the ET product. The associated texts are described below:

" (L354-L366) This surprising result is probably attributed to both the extrapolation issue of ET observations and the high uncertainty of the remote sensing product. Compared with the ensemble simulation of Q (Figure 5(a)) that captures most observed Q, a majority of ET observations exceed the range of the ATS ensemble of ET during the low ET period each year (i.e., wet seasons or September through May next year; see Figure 5(b)). While it is possible that the defined sampling ranges of the two Priestley Taylor coefficients in Table A1 are too limited to provide sufficient variations of ET dynamics, the uncertainty of the MODIS ET product also plays a role here [2, 6]. [6] show that the MODIS ET product has much poorer performance and higher uncertainty in the Colorado Basin than in most of the remaining areas in the United States. The large uncertainty of this remote sensing product probably results from the increasing error in the satellite data caused by the cloudier sky in the mountainous region [5], particularly during the dry seasons (i.e., May through September) [6]. In other words, although the ET ensemble gives a better coverage on the observations in the dry seasons than the wet seasons (Figure 5(b)), that could be due to the underestimation

of the MODIS ET in the dry period with high ET such that the mismatch between the ET ensemble and the observed ET could be probably more significant. "

> Author name: Should the third author be Alexander?

Dr. Sun's first name is corrected now.

**References**

[1] P. Jiang, K. Son, M. K. Mudunuru, and X. Chen. Using mutual information for global sensitivity analysis on watershed modeling. *Water Resources Research*, 58(10), 2022.

[2] M. S. Khan, U. W. Liaqat, J. Baik, and M. Choi. Stand-alone uncertainty characterization of gleam, gldas and mod16 evapotranspiration products using an extended triple collocation approach. *Agricultural and Forest Meteorology*, 252:256–268, 2018.

[3] H. Kling, M. Fuchs, and M. Paulin. Runoff conditions in the upper danube basin under an ensemble of climate change scenarios. *Journal of Hydrology*, 424-425:264–277, 2012.

[4] J. Nash and J. Sutcliffe. River flow forecasting through conceptual models part i — a discussion of principles. *Journal of Hydrology*, 10(3):282–290, 1970.

[5] G. B. Senay, S. Bohms, R. K. Singh, P. H. Gowda, N. M. Velpuri, H. Alemu, and J. P. Verdin. Operational evapotranspiration mapping using remote sensing and weather datasets: A new parameterization for the sseb approach. *JAWRA Journal of the American Water Resources Association*, 49(3):577–591, 2013.

[6] T. Xu, Z. Guo, Y. Xia, V. G. Ferreira, S. Liu, K. Wang, Y. Yao, X. Zhang, and C. Zhao. Evaluation of twelve evapotranspiration products from machine learning, remote sensing and land surface models over conterminous united states. *Journal of Hydrology*, 578:124105, 2019.

---

## Author Comment (AC2)

**Responses to the Comments of Reviewer #2 on ⟨hess-2022-282⟩**

Peishi Jiang      Pin Shuai      Alexander Sun      Maruti K. Mudunuru

Xingyuan Chen

December 28, 2022

> This study aims at basin scale parameter calibration for a physical hydrologic
> model (ATS) using DL-based inverse method.  The authors leveraged the mutual
> information (MI) for the global sensitivity analysis to identify the relation
> between parameters and model simulations, which was later applied to the input
> selection of a MLP parameter inverse model.  They executed different groups of
> simulations and analyses to comprehensively evaluate the proposed framework.  The
> MS is well-written with overall structure easy to follow.  I provide my suggestions
> below regarding better clarifying several points and hopefully they can be useful
> to further improving the quality of this study.

Thank you for the accurate summary and we appreciate your careful reviews and comments.

> As my understanding on this study, the title ''knowledge-informed DL'' is mainly
> represented by the MI sensitivity analysis used in the input selection for the
> following inverse modeling.  Knowledge informed learning, generally in my mind,
> is applying physical laws or constraints to the data driven model based on our
> domain knowledge.  To bridge the proposed MI and physical processes together and
> better strengthen the headline of this study, I suggest the authors try to link
> the MI results with physical processes of the study area and give some physical
> explanations of the results from sensitivity analysis.  This can further highlight
> the physical representations of this study.

We have added the following description to better delineate the knowledge obtained from the sensitivity analysis which further facilitated the follow up inverse mapping development:

" (L304-L314) **Physical knowledge obtained by MI analysis.** The sensitivity analysis reveals the seasonal importance of these watershed characteristics to the hydrological fluxes in this area (Figure 5). During the low flow period (September through March of next year), Q is mostly controlled by the subsurface permeability (i.e., perm_g1, perm_s3, and perm_s4) which regulates both the infiltration and the groundwater movement. Transpiration also plays a role in driving the low flow dynamics through the Priestley Taylor coefficient (e.g., priestley_taylor_alpha_transpiration). During the high flow period (March through September), the snow melting process turns out to be the most critical factor in contributing to the large runoffs, which complies with the prior knowledge about the dominance of the snow process in this watershed. Likewise, the total ET is by and large attributed to a variety of evaporation and transpiration. Snow evaporation is the main component of the total ET in both late autumn and winter when the snow melting rarely happens. On the other hand, in warmer and high-flow seasons, transpiration becomes the dominant contributor to the total ET. The seasonable pattern of the sensitivity of each parameter not only uncovers the hydrological process in the watershed but also serves as the basis to select the most informative model responses to estimate each model parameter. "

I am still confused at the details about how the inverse framework is set up
and trained.  My understanding is that you first run some simulations with ATS
(how are the parameters first initialized here?)  and use the simulations and
parameters to train an inverse mapping with inputs selected by MI, and then
replace ATS simulations with real observations to estimate parameters.  Does the
\responses" mentioned throughout the paper mean the simulated ATS discharge and
ET? What are the training targets and how do you develop the structure, tune the
hyperparameters and train the DL framework?  What are the training and testing
dataset separation?...  Maybe I didn't understand some parts very well, but indeed
expect the authors can better clarify their methodology and results to make readers
more easily understand this work.

Correct, we generated ensemble simulations of ATS to perform both MI-based global sensitivity analysis and develop the deep learning (DL)-based inverse mappings. The mappings, developed using multilayer perceptrons (MLPs), estimate model parameters from model responses that refer to streamflow and ET. The technical details of DL model development were described in the appendix of the preprint version. For better readability, we now moved it to Section 2.4 of the main manuscript and revised the associated texts as follows:

" (L240-L265) For comparison purposes, we developed both the original inverse mapping and our proposed knowledge-informed version for parameter estimation. While a separate neural network is developed for estimating each parameter by using knowledge-informed inverse mapping (Figure 3(b)), the original inverse mapping estimates all parameters using one neural network and is developed by following the same strategy in [1] and [3] (Figure 3(a)). Further, to assess the impact of different responses in calibration, we developed three types of inverse mappings that take various model responses: (1) using both Q and ET; (2) using only Q; and (3) using only ET. Additionally, a multi-year analysis was performed by training inverse mappings using Q of different combinations of observed years to evaluate both the impacts of the dry versus wet years and the number of observed years used in calibration.

All the inverse mappings developed in this study are listed in Table 1. Each mapping was developed using a multilayer perceptron (MLP) model as follows. The input of an MLP is an array concatenating the responses to be assimilated within a given calibration period. The output is the model parameter(s). Let's denote the number of input neurons, output neurons, and hidden layers as $N_i$, $N_o$, and $N_l$, respectively. $N_i$ depends on the type of inverse mapping (with or without being knowledge guided), the selections of the response variable(s), and the number of calibration years, varying from $\sim$100 using one year of Q to 1,785 using all three years of Q and ET. $N_o$ equals either one (i.e., estimating each parameter using knowledge-informed DL calibration) or the number of all the parameters (i.e., using inverse mapping without mutual information). Given $N_i$, $N_o$, and $N_l$, we adopt the arithmetic sequence to determine the number of neurons at each hidden layer $N_{h,l} = \lfloor N_i - \frac{N_i - N_o}{N_l} \times l \rfloor$ (where $1 \leq l \leq N_l$ and $\lfloor \bullet \rfloor$ is the floor function). In doing so, the information from a sequence of observed responses can be gradually propagated to estimate the parameters. We use the leaky ReLu as the nonlinear activation at the end of each layer. Based on the order of the Sobol sequences, we sequentially split the 396 realizations into 300/50/46 for train/validation/test sets, respectively, such that each set is able to cover the full range of the parameter ensemble as much as possible. We trained each MLP using mean square error (MSE) as the loss function over 1,000 epochs with a batch size of 32. The Adam optimization algorithm, a stochastic gradient descent approach, was used to train the neural network. We performed hyperparameter tuning on each MLP using grid search to find the optimal result by varying the number of hidden layers $N_l = [1, 3, 5, 7, 9, 10]$ and the learning rate $l_r = [1e-5, 1e-4, 1e-3]$. The performances of these mappings are further evaluated on the two magnitude-independent metrics, NSE and mKGE. To have consistent comparisons between mappings with and without being knowledge guided, both metrics are computed for the estimation of each parameter based on the test dataset. "

> I didn't understand the result of Figure 7 well and hope the authors can give
> more explanations.  Which variables are the NSE and mKGE calculated on, estimated
> parameters or model simulations?  If they are simulation metric, are theses
> simulations from the model forwarding with parameters estimated from real
> observations (Q & MODIS ET inverse)?  For each individual parameter evaluation,
> how do you set up the values of other parameters when doing ATS forwarding.  The
> caption notifies the performance is reported on testing data, but I didn't see how
> the authors divide testing and training data.

The result of Figure 7 (now Figure 8) was calculated between the true parameters and the estimation by inverse mappings in the test dataset. How the train/validation/test data were splitted is described in the reply to the previous comment regarding the details of DL model development. We revised the caption of Figure 8 as: "Parameter estimation performance of the developed deep learning (DL) inverse mappings on the test dataset using the model responses in the calibration period with regards to (a) the modified Kling-Gupta Efficiency (mKGE) and (b) the Nash-Sutcliffe Efficiency (NSE). Green and light blue represent the mappings without and with being knowledge informed, respectively. Blank, cross, and circle textures are used to represent the mapping using discharge only (qonly), evapotranspiration only (etonly), and both (qet), respectively.".

> I am thinking this multiple-years training VS one-year training discussed in
> section 3.3.  As for multiple years, you choose to increase the input neuron number,
> or keep the one-year structure not changed and just use multiple years data as
> more training samples?  I think the latter one could be more beneficial because
> inputting three-year time series once to the model would require large amounts
> of parameters in the input layer which can be inefficient and overfitted to small
> training data.

For multiple years, we increased the number of inputs of the DL model and performed hyperparameter tuning to find the optimal architecture of the model. The tuning result partially addressed the overfitting issue. Indeed, we observe a limited impact of overfitting from the training result (see Figures 7, A3, and A4). Therefore, we did not try a different model architecture which complicates the hyperparameter tuning procedure and is out of the scope of this study. We have added the following in the result section to demonstrate this point:

" (L316-325) The developed inverse mappings demonstrate limited overfitting issues. Figure 7 plots the training and validation loss over epochs of the seven parameters, each of which is estimated by the knowledge-informed inverse mapping using the corresponding three years of sensitive streamflows (i.e., mi-qonly-3yrs). It can be observed from the figure that both losses quickly decrease with epochs with little discrepancies. Particularly, the parameters sharing with higher mutual information with streamflows show faster convergences of the loss function and do not have overfitting problem (e.g., perm_s3 and snowmelt_degree_diff; see Figure 6(a)). The discrepancy between training and validation losses gets slightly larger for less sensitive parameters (e.g., perm_g4) where streamflow is less informative in parameter estimation. Indeed, informative model responses can provide better parameter estimations, thus reducing the overfitting impact. The limited impact of overfitting is also evident from the NSE and mKGE barplots of the training, validation, and test sets of all the inverse mappings (see Figures A3 and A4), where most mappings have similar performances on parameter estimations among the three sets. "

> Another point I would be interested in is whether the authors have tried
> adding meteorological forcings to the inputs of inverse modeling.  I feel the
> forcing-hydrologic response pair is very important to inform the characteristics
> of basin processes reflected in model parameters.  I am expecting the paired input
> may bring more benefits to this study.

Including atmospheric forcing in the inputs of the neural network might be inappropriate for this basin-specific study. This is because the forcings do not change with ensemble realizations and thus are constant values to the DL model inputs, which might even deteriorate the DL model performance. Including such basin characteristics would be more beneficial to studies encompassing multiple basins. We thus added it as one future work in the conclusion:

" (L458-L460) One potential future work is to develop a unified inverse modeling framework for multiple basins, where the atmospheric forcings and basin characteristics can be also used as the inputs of the inverse mappings in addition to the realization-dependent model responses. "

> Line 76 Do you intend to discuss the overfitting problem here?  Large number of
> weights and limited realizations as training data may cause overfitting with a
> complicated model.

We now discuss the overfitting problem as follows:

" (L77-L79) Further, when using all observed responses as inputs, the potentially large amount of trainable weights of the DL model can make the model training hard and cause the overfitting of the model [4], thus calling for more realizations used in training. "

> Line 177 Please also give explanations for H(Y|X) to help readers' understanding.

We have added the explanation for $H(Y|X)$ in L184-L185: "$H(Y|X)$ is the conditional entropy that quantifies the uncertainty of $Y$ given the knowledge of $X$".

> Line 258 and 259 How did the authors safely draw the conclusion of ''improves
> the MI estimations'' and ''the parameters are falsely considered'' based on the
> differences of preliminary and full analysis?

This is due to the improved MI estimation of the fully analysis which uses around 400 realizations, as evidenced by the convergence of the MI estimations shown in Figure A2 of the appendix. This converged MI estimation allows us to identify the that is not available in the preliminary analysis. We have revised the associated text to better illustrate this point in:

" (L293-L301) By using more realizations, this complete MI analysis shows a better delineation of parameter sensitivity than the preliminary analysis due to its convergence on MI estimation (see the convergence of the parameter rankings in Figure A2). The convergence on a few hundred realizations is consistent with another MI-based sensitivity analysis study using Soil & Water Assessment Tool (SWAT) [2]. Further, the MI-based parameter ranking suggests that compared with the preliminary analysis, the full analysis (1) improves the MI estimations (e.g., perm_s3); and (2) identifies the insensitive parameters (e.g., perm_s4) that are falsely considered sensitive due to the limited samples in the preliminary analysis (see Figure 6). The main permeability in the soil layer (i.e., perm_s3), for example, now shows higher and more temporally coherent sensitivity to Q (Figure 5(a)). On the other hand, perm_s4, which shows some sensitivity in the preliminary analysis, turns out to be insensitive to both Q and ET with almost zero MI at each time step. "

> Additionally, is it possible that in the preliminary analysis some parameters are
> not identified but actually behave sensitive if you include them in the full MI
> analysis?

Yes, it is possible, because the preliminary analysis does not theoretically exclude such false negative cases due to the limited sampling. However, the statistical significance test used to filter the insignificant MI estimation can greatly improve the MI estimation as shown in a previous study in [2], thus partially eliminating such cases. We acknowledge this point in the conclusion:

" (L433-L439) The proposed hierarchical way of sensitivity analysis efficiently utilizes the available limited computational resource through a combination of a prescreening analysis and then a full analysis. Although the prescreening using 50 model runs does not theoretically exclude a false negative case that a sensitive parameter is classified as insensitive, the statistical significance test is able to improve the estimation of mutual information in Figure 4 thus facilitating narrowing down an "accurate" list of parameters to be estimated. Based on the shortened parameter list, a full sensitivity analysis is successfully performed using nearly 400 model runs and provides physically meaningful results on the dependency between the parameters and model responses in Figure 5. "

> Figure 8 The inputs to the inverse model here are real observations or simulated
> responses?

The forward runs (now shown in Figures 9 and 10) are driven by the parameters estimated by the observations. We have revised the captions of the two figures accordingly.

**References**

[1] E. Cromwell, P. Shuai, P. Jiang, E. T. Coon, S. L. Painter, J. D. Moulton, Y. Lin, and X. Chen. Estimating watershed subsurface permeability from stream discharge data using deep neural networks. *Frontiers in Earth Science*, 9, 2021.

[2] P. Jiang, K. Son, M. K. Mudunuru, and X. Chen. Using mutual information for global sensitivity analysis on watershed modeling. *Water Resources Research*, 58(10), 2022.

[3] M. K. Mudunuru, K. Son, P. Jiang, and X. Chen. Swat watershed model calibration using deep learning, 2021.

[4] X. Ying. An overview of overfitting and its solutions. *Journal of Physics: Conference Series*, 1168(2):022022, feb 2019.

---

## Author Comment (AC3)

**Responses to the Comments of Reviewer #3 on ⟨hess-2022-282⟩**

Peishi Jiang     Pin Shuai     Alexander Sun     Maruti K. Mudunuru
Xingyuan Chen

December 28, 2022

> General comments:
>     This paper proposes a knowledge-informed deep learning method that can reduce
> the computational demand required by the calibration of the computationally
> expensive environmental model.  I like the proposed MI sensitivity analysis best
> because it is able to disclose the sensitivity of parameters varying along with
> time which traditional sensitivity analysis is not capable of.  Please see the
> comments in the attached PDF file for suggestions and questions.

Thank you for reviewing our manuscript. We made modifications per the comments below.

> Line 13:  ''all observations'' --> observations covering all time steps

Revised.

> Line 137:  ''continuous discharge (Q)'' --> observed daily

Revised.

> Line 214:  When this inverse mapping is trained, how to select the significant time
> steps from all time steps for each parameter?  The union of the time steps that are
> significant in using Q only and using ET only?

The significant modeled responses (either Q or ET) are identified prior to the development of a knowledge-guided inverse mapping that uses these responses as the inputs. A response at a time step is considered as significant to a parameter if its corresponding mutual information is non-zero based on a statistical significance test, which is described below:

"(L190-L193)In this study, we follow a similar strategy of [2] to estimate $p$ using 10 evenly divided bins along each dimension and perform SST tests to filter out any non-significant MI value with a significance level of 95% based on 100 bootstrap samples. In other words, the computed MI is set to zero if the statistical significance test fails."

In the case of using both Q and ET to estimate a parameter, we took those Q and ET that have non-zero mutual information and concatenated them into an array to the inputs of the knowledge-guided inverse mapping (as elaborated in Figure 3(b)).

> Lines 225-226: It's not clear what the epsilon is because there is no definition
> of or equation defining the noise and observation error.
>     I assume what you mean by the observation error is the standard deviation of
> the observation, and epsilon is 1/3 of observation standard deviation. I am not
> sure whether my understanding is right.

To clarify, we modified the associated sentences as follows:

" (L226-L230) To this end, we generate 100 realizations of noisy observations, denoted as $\mathbf{o}_n$, such that $\mathbf{o}_n = \mathbf{o} + \epsilon \times \mathbf{o} \times \mathbf{r}$, where $\mathbf{o}$ is the vector of the original observations, $\mathbf{r}$ is the random vector with the same size as $\mathbf{o}$ and is drawn from a standard normal distribution, and $\epsilon$ is the standard deviation of the random vector $\mathbf{r}$ and is usually taken as $1/3$ of a given observation error. Following [1], $\epsilon$ is set to 0.0166 for a 5% observation error in this study. "

> Line 273: ''redish'' --> greenish

Revised.

> Without reading Cromwell et al (2021) and Mudunuru et al (2021) about the original
> inverse mapping that estimates all parameters from all responses(illustrated in
> Figure 3(a)), I have no idea why the NSEs using the same response (for example Q)
> for different parameters are different. To my understanding, the test data (46
> out of 396 realizations including all time steps instead of selected time steps)
> are the same in each case (q, et or qet) for different parameters in the original
> inverse mapping since all the parameters are estimated together. Hence, the NSE
> using the same response (for example Q) for different parameters should be the same.
> How are the NSEs using data from the original inverse calculated for each of the
> parameters in the three cases?
>     To make the complicated figure easy to digest, you might consider to remove 7(a)
> in the upper pannel.

We calculated NSE and mKGE for the estimation of each parameter, instead of all the parameters. The purpose is to compare the parameter estimation by the original and the knowledge-guided mappings. We clarify this point in the following sentence:

" (L263-L265) The performances of these mappings are further evaluated on the two magnitude-independent metrics, NSE and mKGE. To have consistent comparisons between mappings with and without being knowledge guided, both metrics are computed for the estimation of each parameter based on the test dataset. "

> Could you add the default ATS run which uses the default parameter values instead
> of the estimated parameter values in Figure 8 and Figure 9?

In order not to complicate the two figures, we now add a figure in the Appendix (Figure A7) that compares the default and the calibrated ATS runs, showing the improvement of the model performance using the knowledge-informed inverse mapping.

> Lines 345-347: I have no idea what this sentence is for.

This sentence is used to indicate the importance of the discharge fluctuations during the low flow period of the dry year in model calibration. We revised the sentence as follows:

" (L) Our finding on the significance of dry year discharge in model calibration indirectly supports some recent studies. [3] found that high flow provides limited information to calibrate models in snow-dominated catchments. This is mainly because there are fewer discharge fluctuations during snow melting or high flow period than rainfall-fed catchments [4]. The decreased role of high flow, in turn, enhances the importance of the low flow period in calibration, particularly in dry years. Indeed, in this watershed, we do observe stronger diurnal discharge fluctuations during the low flow period of the dry year (i.e., WY2018) than the other two wetter years (see Figure A3 in the appendix), which facilitates the better calibration result using observations from the dry year. "

> Figure 12: Could you please plot the observation in a different color? Maybe red?

Figure 12 (now Figure 13) is updated now.

> Line 388: It's not clear to me what the number of input means. Does it mean the total time steps of the selected time steps? Please indicate the time step of observed Q and modeled Q from ATS in section 2.1 and 2.2.

The input refers to the input neurons of a neural network. For knowledge-informed inverse mapping, the input is an array concatenating the responses (i.e., with non-zero mutual information) to be assimilated within a given calibration period. So, the number of inputs is the number of these selected responses used for parameter estimation. We revised the associated texts as below:

" (L248-L253) Each mapping was developed using a multilayer perceptron (MLP) model as follows. The input of an MLP is an array concatenating the responses to be assimilated within a given calibration period. The output is the model parameter(s). Let's denote the number of input neurons, output neurons, and hidden layers as $N_i$, $N_o$, and $N_l$, respectively. $N_i$ depends on the type of inverse mapping (with or without being knowledge guided), the selections of the response variable(s), and the number of calibration years, varying from ~100 using one year of Q to 1,785 using all three years of Q and ET. "

> Line 394: So the Adam optimization is used to tune the values of the number of hidden layers Nl and the learning rate within the sets listed below?

The Adam algorithm is a stochastic gradient descent approach to optimize the parameters of a deep learning model and is used for each MLP development. The hyperparameter tuning was done through a grid search to find the optimal hyperparameters, where each trial/training employs the Adam algorithm to optimize the loss. The related sentences are revised as follows:

" (L259-263) We trained each MLP using mean square error (MSE) as the loss function over 1,000 epochs with a batch size of 32. The Adam optimization algorithm, a stochastic gradient descent approach, was used to train the neural network. We performed hyperparameter tuning on each MLP using grid search to find the optimal result by varying the number of hidden layers $N_l = [1, 3, 5, 7, 9, 10]$ and the learning rate $l_r = [1e-5, 1e-4, 1e-3]$. "

**References**

[1] E. Cromwell, P. Shuai, P. Jiang, E. T. Coon, S. L. Painter, J. D. Moulton, Y. Lin, and X. Chen. Estimating watershed subsurface permeability from stream discharge data using deep neural networks. *Frontiers in Earth Science*, 9, 2021.

[2] P. Jiang, K. Son, M. K. Mudunuru, and X. Chen. Using mutual information for global sensitivity analysis on watershed modeling. *Water Resources Research*, 58(10), 2022.

[3] S. Pool, D. Viviroli, and J. Seibert. Value of a limited number of discharge observations for improving regionalization: A large-sample study across the united states. *Water Resources Research*, 55(1):363–377, 2019.

[4] D. Viviroli and J. Seibert. Can a regionalized model parameterisation be improved with a limited number of runoff measurements? *Journal of Hydrology*, 529:49–61, 2015.

---

## Referee Report (RR1)

**Responses to the Comments of Reviewer #1 on ⟨hess-2022-282⟩**

Peishi Jiang        Pin Shuai        Alexander Sun        Maruti K. Mudunuru

Xingyuan Chen

January 8, 2023

**Comments by Reviewer #1**

> General comments:
>     This study showcases a deep learning optimization method for a high-resolution
> hydrologic model supported by information theory.  I appreciate the honest
> evaluation of the methodology, in-depth reasoning of the deteriorating model
> performance for ET, and examination of results and conclusions aligned with
> earlier studies.  In general, this paper is well-written with a novel contribution.
> However, I think the paper would be stronger if the authors can address the
> following comments.

Thank you for the thorough review of our manuscript. We addressed each comment as shown below.

> Model validation for climate sensitivity:  Currently, the model validation period
> overlaps with the period for calibrating ATS parameters.  I am curious whether
> the optimized parameters would be able to capture the climate sensitivity on flow
> and ET, i.e., improving the flow/ET performance outside of the calibrating period
> (2016-2019).  It would strongly support this tool's eligibility in climate change
> studies.

We now extend the simulation period to 31-12-2021, which is the end of the available Daymet forcing (see Figure 1(b)). We split the whole period into the calibration (1-10-2016 through 30-9-2019) and the evaluation (1-10-2019 through 31-12-2021) periods, used to calibrate and evaluate the climate sensitivity of the model, separately. The result in Figures 9 and 10 shows that the performances of the calibrated ATS during the evaluation period are very close to that of the calibration period, suggesting the adaptability of the estimated parameters to an uncalibrated time period. We revised the associated results and discussion as follows:

" (L232-L239) Here, we separate the entire observations in Figure 1(b) into model calibration and evaluation periods in order to assess the adaptability of the estimated parameters to an uncalibrated period. To this end, we calibrate ATS only using the simulations during water year 2017 to water year 2019 and used the remaining observations (till 31 December 2021) for model evaluation. The ensemble runs used for sensitivity analysis and inverse modeling are performed during the calibration period. The calibrated ATS forward runs were then performed on both periods and compared against the observations in Figure 1(b). We assess the performances of the calibrated models on both periods by using two scale-independent metrics: the Nash-Sutcliffe Efficiency (NSE; [6]) and the modified Kling-Gupta Efficiency (mKGE; [4]).

...

(L346-L352) **Adaptability of the calibrated model in the evaluation period.** For both Q and ET, both NSE and mKGE of the evaluation period (the cyan lines) are astonishingly close, if not identical, to

that of the calibration period (the blue lines). Whenever the calibrated model shows improvement using the knowledge-informed inverse mapping (such as the comparison between qonly-3yrs and mi-qonly-3yrs), we can observe the corresponding improvement in the evaluation period. Such consistent performance between the two periods suggests the robustness of the estimated parameters to climate sensitivity. "

> ET from flux tower: In this study, the authors have demonstrated that worse ET
> performance results from poor quality of MODIS ET products. In this study region,
> is there ET data from the flux tower that could be used for implementing this
> workflow? Even though the flux tower ET data has less spatial coverage, the data
> quality can be better, which might be more useful than MODIS ET when calibrating
> hydrologic parameters.

We looked into the AmeriFlux and there is no flux tower site available in this watershed. Therefore, we are not able to perform the calibration against the site-based observations.

> Specific comments:
>     L158: Can the authors elaborate on what five soil types and four geological
> types are?

They are grouped subsurface characteristics in the soil and geological layers using k-means clustering. We add the following description for a better elaboration (L161-L163): "Each clustered soil or geological type is associated with a specific set of subsurface characteristics (such as permeability), which are assigned to the corresponding grouped grid cells. These subsurface characteristics are important in controlling flow dynamics and can be estimated from hydrological observations."

> L160: A 1000-year spin-up is extremely long. Can the authors briefly explain the
> reason for this long spin-up even if it might be explained in Shuai et al 2022?

We have revised the associated text to clearly explain the motivation for the 1000-year cold or steady-state spinup (note that the cold spin-up took less than one hour to complete on 128 CPU cores due to the faster model convergence once it reached quasi-steady-state.): " (L163-L167) To ensure that the model achieved a physically appropriate initial state, two spinups were performed sequentially, including (1) a cold spinup that ran the model for 1000 years by using constant rainfall and led to steady-state condition at the end of the simulation (e.g., converged total amount of subsurface water storage) and (2) a warm spinup that was initialized by the steady-state spinup result and performed a transient simulation for 10 years (i.e., 1 October 2004 – 1 October 2014) under the Daymet forcing. "

> L162: Could the authors briefly explain how they preselected the parameters in
> this study?
>     L208: Does the MI have to be zero? If the MI between a parameter and the
> model responses is small enough, is it possible to neglect that parameter? What
> would be a proper threshold for it?
>     L249-250: Given the narrowed list, it seems that the authors eliminated the
> parameters with small MI (not zero), which slightly contradicts the previous
> statement where only parameters with zero MI would be eliminated (L208). It
> would be helpful to clarify the threshold of MI below which the parameters will
> be eliminated.

As all three comments are associated with how we performed the preliminary sensitivity analysis using mutual information, we reply to them in one thread here. In short, the preselection is based on the mutual information (MI) computed for each parameter and each response at a given time step. For a given model response (e.g., Q), we say it is sensitive to a parameter if the proportion of non-zero MI over all the time steps is greater than a given threshold (i.e., 5% in this study). For each MI calculation, we performed a statistical significance test to determine whether the computed MI is significant and set MI to zero if the test fails. So, the MI can be zero. We enriched the description in the associated texts as below:
"

" (L190-L194) In this study, we follow a similar strategy of [2] to estimate $p$ using 10 evenly divided bins along each dimension and perform SST tests to filter out any non-significant MI value with a significance level of 95% based on 100 bootstrap samples. In other words, the computed MI is set to zero if the statistical significance test fails.

...

(L205-L209) This preliminary MI analysis would allow filtering out the parameters that show little sensitivity to the model responses, thus reducing the number of parameters to be calibrated. This filtering process is performed based on whether a parameter demonstrates sufficient sensitivity across the simulation period. In this study, we selected the parameter whose proportion of the non-zero MI is larger than 5% of the overall time steps for the following full sensitivity analysis.

...

(L285-L290) Based on the proportion of nonzero MI over all the time steps (see Figure A1 in the appendix), we find that Q is mostly sensitive to (using a threshold of 5%) perm_s3, perm_s4, perm_g1, perm_g4, snowmelt_rate, snowmelt_degree_diff, and priestley_taylor_alpha_transpiration, and ET is mostly sensitive to priestley_taylor_alpha_transpiration, priestley_taylor_alpha_snow, perm_s3, perm_g1, and perm_g4. Consequently, we narrow down the parameters to be calibrated by taking the union of the two sets of parameters that show sensitivities to either Q or ET (also highlighted in Table A1). "

> L208-210:  Interesting!  Great summary!

Thank you for the generous comment.

> L215:  When training using different combinations of years, why do the authors
> only look at Q, not ET?

We do not use ET for multi-year analysis because the extrapolation issue of the ET observations deteriorates the parameter estimations using the inverse mapping, as described in Section 3.3. In other words, a multi-year analysis including ET would be questionable and not trustworthy to evaluate the impact of dry and wet years. Therefore, we performed the multi-year analysis against only Q.

> L286-287:  Please clarify whether the extrapolation issue partially or solely
> contributes to the worse MI-informed results.

The inferior calibrated ATS runs using knowledge-informed deep learning are attributed to both the extrapolation issue of the observations and the potential high uncertainty of the ET product. The associated texts are described below:
" (L354-L366) This surprising result is probably attributed to both the extrapolation issue of ET observations and the high uncertainty of the remote sensing product. Compared with the ensemble simulation of Q (Figure 5(a)) that captures most observed Q, a majority of ET observations exceed the range of the ATS ensemble of ET during the low ET period each year (i.e., wet seasons or September through May next year; see Figure 5(b)). While it is possible that the defined sampling ranges of the two Priestley Taylor coefficients in Table A1 are too limited to provide sufficient variations of ET dynamics, the uncertainty of the

MODIS ET product also plays a role here [3, 10]. [10] show that the MODIS ET product has much poorer performance and higher uncertainty in the Colorado Basin than in most of the remaining areas in the United States. The large uncertainty of this remote sensing product probably results from the increasing error in the satellite data caused by the cloudier sky in the mountainous region [8], particularly during the dry seasons (i.e., May through September) [10]. In other words, although the ET ensemble gives a better coverage on the observations in the dry seasons than the wet seasons (Figure 5(b)), that could be due to the underestimation of the MODIS ET in the dry period with high ET such that the mismatch between the ET ensemble and the observed ET could be probably more significant. "

> `Author name:  Should the third author be Alexander?`

Dr. Sun's first name is corrected now.

**Comments by Reviewer #2**

> This study aims at basin scale parameter calibration for a physical hydrologic
> model (ATS) using DL-based inverse method. The authors leveraged the mutual
> information (MI) for the global sensitivity analysis to identify the relation
> between parameters and model simulations, which was later applied to the input
> selection of a MLP parameter inverse model. They executed different groups of
> simulations and analyses to comprehensively evaluate the proposed framework. The
> MS is well-written with overall structure easy to follow. I provide my suggestions
> below regarding better clarifying several points and hopefully they can be useful
> to further improving the quality of this study.

Thank you for the accurate summary and we appreciate your careful reviews and comments.

> As my understanding on this study, the title ''knowledge-informed DL'' is mainly
> represented by the MI sensitivity analysis used in the input selection for the
> following inverse modeling. Knowledge informed learning, generally in my mind,
> is applying physical laws or constraints to the data driven model based on our
> domain knowledge. To bridge the proposed MI and physical processes together and
> better strengthen the headline of this study, I suggest the authors try to link
> the MI results with physical processes of the study area and give some physical
> explanations of the results from sensitivity analysis. This can further highlight
> the physical representations of this study.

We have added the following description to better delineate the knowledge obtained from the sensitivity
analysis which further facilitated the follow up inverse mapping development:

" (L304-L314) **Physical knowledge obtained by MI analysis.** The sensitivity analysis reveals
the seasonal importance of these watershed characteristics to the hydrological fluxes in this area (Figure
5). During the low flow period (September through March of next year), Q is mostly controlled by the
subsurface permeability (i.e., perm_g1, perm_s3, and perm_s4) which regulates both the infiltration and
the groundwater movement. Transpiration also plays a role in driving the low flow dynamics through the
Priestley Taylor coefficient (e.g., priestley_taylor_alpha_transpiration). During the high flow period (March
through September), the snow melting process turns out to be the most critical factor in contributing to
the large runoffs, which complies with the prior knowledge about the dominance of the snow process in this
watershed. Likewise, the total ET is by and large attributed to a variety of evaporation and transpiration.
Snow evaporation is the main component of the total ET in both late autumn and winter when the snow
melting rarely happens. On the other hand, in warmer and high-flow seasons, transpiration becomes the
dominant contributor to the total ET. The seasonable pattern of the sensitivity of each parameter not only
uncovers the hydrological process in the watershed but also serves as the basis to select the most informative
model responses to estimate each model parameter. "

> I am still confused at the details about how the inverse framework is set up
> and trained.  My understanding is that you first run some simulations with ATS
> (how are the parameters first initialized here?)  and use the simulations and
> parameters to train an inverse mapping with inputs selected by MI, and then
> replace ATS simulations with real observations to estimate parameters.  Does the
> \responses" mentioned throughout the paper mean the simulated ATS discharge and
> ET? What are the training targets and how do you develop the structure, tune the
> hyperparameters and train the DL framework?  What are the training and testing
> dataset separation?...  Maybe I didn't understand some parts very well, but indeed
> expect the authors can better clarify their methodology and results to make readers
> more easily understand this work.

Correct, we generated ensemble simulations of ATS to perform both MI-based global sensitivity analysis and develop the deep learning (DL)-based inverse mappings. The mappings, developed using multilayer perceptrons (MLPs), estimate model parameters from model responses that refer to streamflow and ET. The technical details of DL model development were described in the appendix of the preprint version. For better readability, we now moved it to Section 2.4 of the main manuscript and revised the associated texts as follows:

" (L240-L265) For comparison purposes, we developed both the original inverse mapping and our proposed knowledge-informed version for parameter estimation. While a separate neural network is developed for estimating each parameter by using knowledge-informed inverse mapping (Figure 3(b)), the original inverse mapping estimates all parameters using one neural network and is developed by following the same strategy in [1] and [5] (Figure 3(a)). Further, to assess the impact of different responses in calibration, we developed three types of inverse mappings that take various model responses: (1) using both Q and ET; (2) using only Q; and (3) using only ET. Additionally, a multi-year analysis was performed by training inverse mappings using Q of different combinations of observed years to evaluate both the impacts of the dry versus wet years and the number of observed years used in calibration.

All the inverse mappings developed in this study are listed in Table 1. Each mapping was developed using a multilayer perceptron (MLP) model as follows. The input of an MLP is an array concatenating the responses to be assimilated within a given calibration period. The output is the model parameter(s). Let's denote the number of input neurons, output neurons, and hidden layers as $N_i$, $N_o$, and $N_l$, respectively. $N_i$ depends on the type of inverse mapping (with or without being knowledge guided), the selections of the response variable(s), and the number of calibration years, varying from $\sim$100 using one year of Q to 1,785 using all three years of Q and ET. $N_o$ equals either one (i.e., estimating each parameter using knowledge-informed DL calibration) or the number of all the parameters (i.e., using inverse mapping without mutual information). Given $N_i$, $N_o$, and $N_l$, we adopt the arithmetic sequence to determine the number of neurons at each hidden layer $N_{h,l} = \lfloor N_i - \frac{N_i - N_o}{N_l} \times l \rfloor$ (where $1 \leq l \leq N_l$ and $\lfloor \bullet \rfloor$ is the floor function). In doing so, the information from a sequence of observed responses can be gradually propagated to estimate the parameters. We use the leaky ReLu as the nonlinear activation at the end of each layer. Based on the order of the Sobol sequences, we sequentially split the 396 realizations into 300/50/46 for train/validation/test sets, respectively, such that each set is able to cover the full range of the parameter ensemble as much as possible. We trained each MLP using mean square error (MSE) as the loss function over 1,000 epochs with a batch size of 32. The Adam optimization algorithm, a stochastic gradient descent approach, was used to train the neural network. We performed hyperparameter tuning on each MLP using grid search to find the optimal result by varying the number of hidden layers $N_l = [1, 3, 5, 7, 9, 10]$ and the learning rate $l_r = [1e-5, 1e-4, 1e-3]$. The performances of these mappings are further evaluated on the two magnitude-independent metrics, NSE and mKGE. To have consistent comparisons between mappings with and without being knowledge guided, both metrics are computed for the estimation of each parameter based on the test dataset. "

> I didn't understand the result of Figure 7 well and hope the authors can give
> more explanations.  Which variables are the NSE and mKGE calculated on, estimated
> parameters or model simulations?  If they are simulation metric, are theses
> simulations from the model forwarding with parameters estimated from real
> observations (Q & MODIS ET inverse)?  For each individual parameter evaluation,
> how do you set up the values of other parameters when doing ATS forwarding.  The
> caption notifies the performance is reported on testing data, but I didn't see how
> the authors divide testing and training data.

The result of Figure 7 (now Figure 8) was calculated between the true parameters and the estimation by inverse mappings in the test dataset. How the train/validation/test data were splitted is described in the reply to the previous comment regarding the details of DL model development. We revised the caption of Figure 8 as: "Parameter estimation performance of the developed deep learning (DL) inverse mappings on the test dataset using the model responses in the calibration period with regards to (a) the modified Kling-Gupta Efficiency (mKGE) and (b) the Nash-Sutcliffe Efficiency (NSE). Green and light blue represent the mappings without and with being knowledge informed, respectively. Blank, cross, and circle textures are used to represent the mapping using discharge only (qonly), evapotranspiration only (etonly), and both (qet), respectively.".

> I am thinking this multiple-years training VS one-year training discussed in
> section 3.3.  As for multiple years, you choose to increase the input neuron number,
> or keep the one-year structure not changed and just use multiple years data as
> more training samples?  I think the latter one could be more beneficial because
> inputting three-year time series once to the model would require large amounts
> of parameters in the input layer which can be inefficient and overfitted to small
> training data.

For multiple years, we increased the number of inputs of the DL model and performed hyperparameter tuning to find the optimal architecture of the model. The tuning result partially addressed the overfitting issue. Indeed, we observe a limited impact of overfitting from the training result (see Figures 7, A3, and A4). Therefore, we did not try a different model architecture which complicates the hyperparameter tuning procedure and is out of the scope of this study. We have added the following in the result section to demonstrate this point:

" (L316-325) The developed inverse mappings demonstrate limited overfitting issues. Figure 7 plots the training and validation loss over epochs of the seven parameters, each of which is estimated by the knowledge-informed inverse mapping using the corresponding three years of sensitive streamflows (i.e., mi-qonly-3yrs). It can be observed from the figure that both losses quickly decrease with epochs with little discrepancies. Particularly, the parameters sharing with higher mutual information with streamflows show faster convergences of the loss function and do not have overfitting problem (e.g., perm_s3 and snowmelt_degree_diff; see Figure 6(a)). The discrepancy between training and validation losses gets slightly larger for less sensitive parameters (e.g., perm_g4) where streamflow is less informative in parameter estimation. Indeed, informative model responses can provide better parameter estimations, thus reducing the overfitting impact. The limited impact of overfitting is also evident from the NSE and mKGE barplots of the training, validation, and test sets of all the inverse mappings (see Figures A3 and A4), where most mappings have similar performances on parameter estimations among the three sets. "

> Another point I would be interested in is whether the authors have tried
> adding meteorological forcings to the inputs of inverse modeling.  I feel the
> forcing-hydrologic response pair is very important to inform the characteristics
> of basin processes reflected in model parameters.  I am expecting the paired input
> may bring more benefits to this study.

Including atmospheric forcing in the inputs of the neural network might be inappropriate for this basin-specific study. This is because the forcings do not change with ensemble realizations and thus are constant values to the DL model inputs, which might even deteriorate the DL model performance. Including such basin characteristics would be more beneficial to studies encompassing multiple basins. We thus added it as one future work in the conclusion:

" (L458-L460) One potential future work is to develop a unified inverse modeling framework for multiple basins, where the atmospheric forcings and basin characteristics can be also used as the inputs of the inverse mappings in addition to the realization-dependent model responses. "

> Line 76 Do you intend to discuss the overfitting problem here?  Large number of
> weights and limited realizations as training data may cause overfitting with a
> complicated model.

We now discuss the overfitting problem as follows:

" (L77-L79) Further, when using all observed responses as inputs, the potentially large amount of trainable weights of the DL model can make the model training hard and cause the overfitting of the model [11], thus calling for more realizations used in training. "

> Line 177 Please also give explanations for H(Y|X) to help readers' understanding.

We have added the explanation for $H(Y|X)$ in L184-L185: "$H(Y|X)$ is the conditional entropy that quantifies the uncertainty of $Y$ given the knowledge of $X$".

> Line 258 and 259 How did the authors safely draw the conclusion of ''improves
> the MI estimations'' and ''the parameters are falsely considered'' based on the
> differences of preliminary and full analysis?

This is due to the improved MI estimation of the fully analysis which uses around 400 realizations, as evidenced by the convergence of the MI estimations shown in Figure A2 of the appendix. This converged MI estimation allows us to identify the that is not available in the preliminary analysis. We have revised the associated text to better illustrate this point in:

" (L293-L301) By using more realizations, this complete MI analysis shows a better delineation of parameter sensitivity than the preliminary analysis due to its convergence on MI estimation (see the convergence of the parameter rankings in Figure A2). The convergence on a few hundred realizations is consistent with another MI-based sensitivity analysis study using Soil & Water Assessment Tool (SWAT) [2]. Further, the MI-based parameter ranking suggests that compared with the preliminary analysis, the full analysis (1) improves the MI estimations (e.g., perm_s3); and (2) identifies the insensitive parameters (e.g., perm_s4) that are falsely considered sensitive due to the limited samples in the preliminary analysis (see Figure 6). The main permeability in the soil layer (i.e., perm_s3), for example, now shows higher and more temporally coherent sensitivity to Q (Figure 5(a)). On the other hand, perm_s4, which shows some sensitivity in the preliminary analysis, turns out to be insensitive to both Q and ET with almost zero MI at each time step. "

> Additionally, is it possible that in the preliminary analysis some parameters are
> not identified but actually behave sensitive if you include them in the full MI
> analysis?

Yes, it is possible, because the preliminary analysis does not theoretically exclude such false negative cases due to the limited sampling. However, the statistical significance test used to filter the insignificant MI estimation can greatly improve the MI estimation as shown in a previous study in [2], thus partially eliminating such cases. We acknowledge this point in the conclusion:

" (L433-L439) The proposed hierarchical way of sensitivity analysis efficiently utilizes the available limited computational resource through a combination of a prescreening analysis and then a full analysis. Although the prescreening using 50 model runs does not theoretically exclude a false negative case that a sensitive parameter is classified as insensitive, the statistical significance test is able to improve the estimation of mutual information in Figure 4 thus facilitating narrowing down an "accurate" list of parameters to be estimated. Based on the shortened parameter list, a full sensitivity analysis is successfully performed using nearly 400 model runs and provides physically meaningful results on the dependency between the parameters and model responses in Figure 5. "

> Figure 8 The inputs to the inverse model here are real observations or simulated
> responses?

The forward runs (now shown in Figures 9 and 10) are driven by the parameters estimated by the observations. We have revised the captions of the two figures accordingly.

**Comments by Reviewer #3**

> General comments:
>     This paper proposes a knowledge-informed deep learning method that can reduce
> the computational demand required by the calibration of the computationally
> expensive environmental model.  I like the proposed MI sensitivity analysis best
> because it is able to disclose the sensitivity of parameters varying along with
> time which traditional sensitivity analysis is not capable of.  Please see the
> comments in the attached PDF file for suggestions and questions.

Thank you for reviewing our manuscript. We made modifications per the comments below.

> Line 13:  ''all observations'' --> observations covering all time steps

Revised.

> Line 137:  ''continuous discharge (Q)'' --> observed daily

Revised.

> Line 214:  When this inverse mapping is trained, how to select the significant time
> steps from all time steps for each parameter?  The union of the time steps that are
> significant in using Q only and using ET only?

The significant modeled responses (either Q or ET) are identified prior to the development of a knowledge-guided inverse mapping that uses these responses as the inputs. A response at a time step is considered as significant to a parameter if its corresponding mutual information is non-zero based on a statistical significance test, which is described below:

"(L190-L193)In this study, we follow a similar strategy of [2] to estimate $p$ using 10 evenly divided bins along each dimension and perform SST tests to filter out any non-significant MI value with a significance level of 95% based on 100 bootstrap samples. In other words, the computed MI is set to zero if the statistical significance test fails."

In the case of using both Q and ET to estimate a parameter, we took those Q and ET that have non-zero mutual information and concatenated them into an array to the inputs of the knowledge-guided inverse mapping (as elaborated in Figure 3(b)).

> Lines 225-226:  It's not clear what the epsilon is because there is no definition
> of or equation defining the noise and observation error.
>     I assume what you mean by the observation error is the standard deviation of
> the observation, and epsilon is 1/3 of observation standard deviation.  I am not
> sure whether my understanding is right.

To clarify, we modified the associated sentences as follows:

" (L226-L230) To this end, we generate 100 realizations of noisy observations, denoted as $\mathbf{o}_n$, such that $\mathbf{o}_n = \mathbf{o} + \epsilon \times \mathbf{o} \times \mathbf{r}$, where $\mathbf{o}$ is the vector of the original observations, $\mathbf{r}$ is the random vector with the same size as $\mathbf{o}$ and is drawn from a standard normal distribution, and $\epsilon$ is the standard deviation of the random vector $\mathbf{r}$ and is usually taken as 1/3 of a given observation error. Following [1], $\epsilon$ is set to 0.0166 for a 5% observation error in this study. "

Revised.

    Without reading Cromwell et al (2021) and Mudunuru et al (2021) about the original
    inverse mapping that estimates all parameters from all responses(illustrated in
    Figure 3(a)), I have no idea why the NSEs using the same response (for example Q)
    for different parameters are different.  To my understanding, the test data (46
    out of 396 realizations including all time steps instead of selected time steps)
    are the same in each case (q, et or qet) for different parameters in the original
    inverse mapping since all the parameters are estimated together.  Hence, the NSE
    using the same response (for example Q) for different parameters should be the same.
    How are the NSEs using data from the original inverse calculated for each of the
    parameters in the three cases?
        To make the complicated figure easy to digest, you might consider to remove 7(a)
    in the upper pannel.

We calculated NSE and mKGE for the estimation of each parameter, instead of all the parameters. The purpose is to compare the parameter estimation by the original and the knowledge-guided mappings. We clarify this point in the following sentence:

" (L263-L265) The performances of these mappings are further evaluated on the two magnitude-independent metrics, NSE and mKGE. To have consistent comparisons between mappings with and without being knowledge guided, both metrics are computed for the estimation of each parameter based on the test dataset. "

    Could you add the default ATS run which uses the default parameter values instead
    of the estimated parameter values in Figure 8 and Figure 9?

In order not to complicate the two figures, we now add a figure in the Appendix (Figure A7) that compares the default and the calibrated ATS runs, showing the improvement of the model performance using the knowledge-informed inverse mapping.

This sentence is used to indicate the importance of the discharge fluctuations during the low flow period of the dry year in model calibration. We revised the sentence as follows:

" (L) Our finding on the significance of dry year discharge in model calibration indirectly supports some recent studies. [7] found that high flow provides limited information to calibrate models in snow-dominated catchments. This is mainly because there are fewer discharge fluctuations during snow melting or high flow period than rainfall-fed catchments [9]. The decreased role of high flow, in turn, enhances the importance of the low flow period in calibration, particularly in dry years. Indeed, in this watershed, we do observe stronger diurnal discharge fluctuations during the low flow period of the dry year (i.e., WY2018) than the other two wetter years (see Figure A3 in the appendix), which facilitates the better calibration result using observations from the dry year. "

    Figure 12:  Could you please plot the observation in a different color?  Maybe
    red?

Figure 12 (now Figure 13) is updated now.

> Line 388:  It's not clear to me what the number of input means.  Does it mean
> the total time steps of the selected time steps?  Please indicate the time step
> of observed Q and modeled Q from ATS in section 2.1 and 2.2.

The input refers to the input neurons of a neural network. For knowledge-informed inverse mapping, the input is an array concatenating the responses (i.e., with non-zero mutual information) to be assimilated within a given calibration period. So, the number of inputs is the number of these selected responses used for parameter estimation. We revised the associated texts as below:

" (L248-L253) Each mapping was developed using a multilayer perceptron (MLP) model as follows. The input of an MLP is an array concatenating the responses to be assimilated within a given calibration period. The output is the model parameter(s). Let's denote the number of input neurons, output neurons, and hidden layers as $N_i$, $N_o$, and $N_l$, respectively. $N_i$ depends on the type of inverse mapping (with or without being knowledge guided), the selections of the response variable(s), and the number of calibration years, varying from $\sim$100 using one year of Q to 1,785 using all three years of Q and ET. "

> Line 394:  So the Adam optimization is used to tune the values of the number of
> hidden layers Nl and the learning rate within the sets listed below?

The Adam algorithm is a stochastic gradient descent approach to optimize the parameters of a deep learning model and is used for each MLP development. The hyperparameter tuning was done through a grid search to find the optimal hyperparameters, where each trial/training employs the Adam algorithm to optimize the loss. The related sentences are revised as follows:

" (L259-263) We trained each MLP using mean square error (MSE) as the loss function over 1,000 epochs with a batch size of 32. The Adam optimization algorithm, a stochastic gradient descent approach, was used to train the neural network. We performed hyperparameter tuning on each MLP using grid search to find the optimal result by varying the number of hidden layers $N_l = [1, 3, 5, 7, 9, 10]$ and the learning rate $l_r = [1e-5, 1e-4, 1e-3]$. "

**References**

[1] E. Cromwell, P. Shuai, P. Jiang, E. T. Coon, S. L. Painter, J. D. Moulton, Y. Lin, and X. Chen. Estimating watershed subsurface permeability from stream discharge data using deep neural networks. *Frontiers in Earth Science*, 9, 2021.

[2] P. Jiang, K. Son, M. K. Mudunuru, and X. Chen. Using mutual information for global sensitivity analysis on watershed modeling. *Water Resources Research*, 58(10), 2022.

[3] M. S. Khan, U. W. Liaqat, J. Baik, and M. Choi. Stand-alone uncertainty characterization of gleam, gldas and mod16 evapotranspiration products using an extended triple collocation approach. *Agricultural and Forest Meteorology*, 252:256–268, 2018.

[4] H. Kling, M. Fuchs, and M. Paulin. Runoff conditions in the upper danube basin under an ensemble of climate change scenarios. *Journal of Hydrology*, 424-425:264–277, 2012.

[5] M. K. Mudunuru, K. Son, P. Jiang, and X. Chen. Swat watershed model calibration using deep learning, 2021.

[6] J. Nash and J. Sutcliffe. River flow forecasting through conceptual models part i — a discussion of principles. *Journal of Hydrology*, 10(3):282–290, 1970.

[7] S. Pool, D. Viviroli, and J. Seibert. Value of a limited number of discharge observations for improving regionalization: A large-sample study across the united states. *Water Resources Research*, 55(1):363–377, 2019.

[8] G. B. Senay, S. Bohms, R. K. Singh, P. H. Gowda, N. M. Velpuri, H. Alemu, and J. P. Verdin. Operational evapotranspiration mapping using remote sensing and weather datasets: A new parameterization for the sseb approach. *JAWRA Journal of the American Water Resources Association*, 49(3):577–591, 2013.

[9] D. Viviroli and J. Seibert. Can a regionalized model parameterisation be improved with a limited number of runoff measurements? *Journal of Hydrology*, 529:49–61, 2015.

[10] T. Xu, Z. Guo, Y. Xia, V. G. Ferreira, S. Liu, K. Wang, Y. Yao, X. Zhang, and C. Zhao. Evaluation of twelve evapotranspiration products from machine learning, remote sensing and land surface models over conterminous united states. *Journal of Hydrology*, 578:124105, 2019.

[11] X. Ying. An overview of overfitting and its solutions. *Journal of Physics: Conference Series*, 1168(2):022022, feb 2019.

---

## Author Response (AR2)

**Responses to the Comments of Reviewer#3 on ⟨hess-2022-282⟩**

Peishi Jiang        Pin Shuai        Alexander Sun        Maruti K. Mudunuru

Xingyuan Chen

June 8, 2023

**Comments by Reviewer #3**

> Just would like to confirm that I understand your answer correctly.  What you mean
> is Q (with non-MI)  ET (with non-MI) where  is the union symbol in math, instead of
> a subset of Q (with non-MI) and a subset of ET (with non-MI) filtered by that both
> Q and ET have non-MI values at the same time, correct?

That is correct. We added the following texts in the manuscript to clarify this:

" (L345-L247) Note that in the case of knowledge-informed inverse mapping using both observations, we take the union of the Q and ET that have non-zero mutual information as the inputs of the neural network. "

> Let me make my question clear.
>     My question is whether in the original mapping, all the parameters
> are estimated as a whole.  i.e., when the training dataset is applied,
> the original mapping outputs the estimates of all parameters (shown in
> Figure 3 (a))?  For example, the estimated values of the seven parameters
> are priestley_taylor_alpha-snow=0.01, priestley_taylor_alpha-snow=0.02,
> snowmelt_rate=0.03, snowmelt_degree_diff=0.04, perm_s3=0.05, perm_g1=0.06, and
> perm_g4=0.07.
>     If you applied this combined values of these parameters into the hydrological
> model, you should get the same time series of discharge (q) for all these seven
> parameters because you applied the same set of the parameter values shown above.
> Or for each parameter, you use the estimated value but use the default values for
> the other six parameters so that for each parameter the discharge (q ) calculated
> in the hydrological model is different and thus the NSEs are different for
> different parameters in Figure 8(b).  Or in the original mapping, each parameter
> is estimated separately instead of all the parameters as a whole set?
>     This is my question.

Thank you for the clarification. The original mapping estimate all seven parameters as a whole. Once trained, we further calculate the NSE and mKGE of every parameter estimated by each original mapping on the test dataset so that we can have consistent performance assessment between the original mapping and the knowledge-informed mapping that estimates each parameter separately. To clarify, we included the following explanation in the manuscript:

" (L266-L269) To have consistent comparisons between mappings with and without being knowledge guided, both metrics are computed for the estimation of each parameter based on the test dataset (note that

while the original mapping estimate all seven parameters as a whole during the training, we calculate the two metrics for each parameter separately during the postprocessing). "